# HARBOR: Hierarchical Abduction with Bayesian Orchestration for Reliable Probability Inference in Large Language Models

## Abstract

A central challenge in large-scale decision-making under incomplete information is the estimation of reliable probabilities. Prior work has employed Large Language Models (LLMs) to generate relevant factors and provide initial, coarse-grained probability estimates. These methods typically utilize an LLM for forward abduction to generate factors, with each factor constrained to two mutually exclusive attributes. A Naïve Bayes model is then trained on combinations of these factors to provide more accurate probabilities. However, this approach often yields a sparse factor space, resulting in "unknown" predictions where the model fails to produce an output. Naively increasing the number of factors to densify the space not only introduces statistical noise but also violates the Naïve Bayes independence assumption, ultimately compromising the stability and reliability of the estimates. To address these limitations, we propose HARBOR, a novel inference framework that orchestrates aggregated Bayesian inference over a hierarchically structured factor space. HARBOR first constructs a dense, structured factor space through iterative generation and hierarchical clustering. It then performs context-aware mapping using retrieval and refinement operations on this hierarchy to reduce "unknown" predictions. Finally, HARBOR extends Naïve Bayes by incorporating a Causal Bayesian Network to model latent dependencies, thereby relaxing the strict independence assumption. Experiments show that HARBOR substantially reduces "unknown" predictions and yields more reliable probabilities than direct LLM baselines, achieving state-of-the-art performance with significantly reduced time and token overhead.

## 1 Introduction

Large language models (LLMs) are increasingly adopted in mission-critical decision-making tasks—ranging from emergency response to infrastructure planning—where accurate, interpretable, and unbiased probability estimates are essential for trustworthy outcomes (McCarthy & Hayes, 1981; Sun et al., 2024). The central challenge is to reliably estimate the conditional probability $P(O_i \mid C)$ for two competing hypotheses, $O_1$ and $O_2$, given a context $C$. This context is typically composed of a high-level scenario, $S_{\text{cen}}$, and a specific downstream condition, $U$. Existing abductive frameworks build a factor space based on the general scenario $S_{\text{cen}}$, but often struggle to connect it to the specific details in $U$.

However, LLMs remain ill-suited for direct probability estimation for two key reasons: (1) they often produce numerical confidence scores that are both inaccurate and overconfident (Sun et al., 2024), and (2) they lack an out-of-the-box interpretable and controllable mechanism for explaining how these estimates are derived (Zhu et al., 2024). As a result, we cannot directly leverage LLMs' decision-making capabilities in large-scale automated scenarios, motivating the need for methods to infer more reliable probabilities from LLM outputs.

Recent methods address this by combining LLM-driven abduction with Bayesian inference. A prominent approach, exemplified by BIRD (Feng et al., 2025), as shown on the left side of Figure 1, uses *forward abduction*, where an LLM first generates a factor and then its mutually exclusive

Figure 1: Illustration of current methods' limitations in decision-making on a cooking scenario. **Left:** forward abduction yields a sparse factor space and "unknown" mappings. **Right:** when a condition activates many factors, naïve factor expansion introduces noise and violates the Naïve Bayes independence assumption; HARBOR resolves both via hierarchical factor-space construction and causal Bayesian modeling.

attributes. This method, however, often produces a sparse factor space, causing the mapping to be empty and leading to "unknown" predictions.

However, indiscriminately enlarging the factor space to boost coverage—as shown on the right of Figure 1—inevitably injects statistical noise and creates spurious dependencies among factors. This undermines the naïve Bayes conditional-independence assumption and distorts the resulting probability estimates (Zhang, 2004; Hand & Yu, 2001). Some works have introduced techniques such as attribute weighting, feature grouping, and structural extensions to mitigate this independence violation (Zaidi et al., 2013; Prabha et al., 2022; He et al., 2025). Other related work has explored different facets of combining LLMs and Bayesian reasoning (Reuter et al., 2025; Qiu et al., 2025; Sgouritsa et al., 2024), but the dual challenges of sparsity and dependency in abductive frameworks persist.

To overcome these limitations, we propose HARBOR (**H**ier**ar**chical Abduction with **B**ayesian **O**rchestration for **R**eliable probability inference), a multi-stage framework that: (1) Iteratively builds a dense factor space from LLM-generated sentences using a bottom-up abduction strategy, and organizes these factors into a two-tier hierarchy via clustering and LLM-guided theming; (2) Implements a context-aware mapping pipeline using hierarchical retrieval; and (3) Constructs both a Naïve Bayes model and a Causal Bayesian Network whose parameters are initialized with LLM-elicited priors, capturing latent dependencies between factors. Results show that HARBOR substantially reduces "unknown" predictions and produces probability estimates that are significantly more calibrated and better aligned with human preferences than state-of-the-art baselines. In summary, our contributions are as follows:

- We design a multi-stage abduction pipeline that iteratively expands the factor space with high-quality factors, substantially reducing "unknown" predictions in downstream inference.
- We integrate a causal Bayesian network to model latent factor dependencies, enhancing probability calibration beyond the naïve Bayes assumption.
- We show that HARBOR achieves state-of-the-art performance on preference-based pairwise evaluation while substantially reducing inference time and token usage.

## 2    RELATED WORK

**Decomposition-Based Reasoning**   Like many advanced reasoning frameworks, HARBOR decomposes complex problems into smaller components. This builds on a rich body of work, from explicitly breaking down questions into procedural steps (Wolfson et al., 2020; Madaan et al., 2021) or generating faithful reasoning chains (Tafjord et al., 2022; Zhou et al., 2022), to the now-widespread Chain-of-Thought (CoT) prompting (Wei et al., 2022). More recent structured approaches aim for greater cognitive plausibility (Yao et al., 2023; Lin et al., 2023), maintain an explicit belief graph (Kassner et al., 2023), or treat the textual reasoning chain itself as a latent variable (Hoffman

et al., 2023). While related, the latent variables in HARBOR's Causal Bayesian Network represent abstract concepts rather than token sequences. The primary distinction of our method, however, is its *proactive* factor-space construction. Instead of decomposing reactively for each query, HARBOR employs "bottom-up abduction" to build a comprehensive, hierarchically structured factor space. This design mitigates the issue of factor sparsity, creating a persistent and reusable reasoning structure that reduces the generation of "unknown" predictions.

**Probabilistic Inference and Uncertainty Estimations**   Many existing efforts aim to improve the uncertainty estimation of LLMs. Some approaches rely on direct model outputs, such as token probabilities (Ren et al., 2023) or verbalized confidence scores (Xiong et al., 2023), which are often poorly calibrated. Others employ sampling-based techniques (Kuhn et al., 2023) or fine-tune the model to amortize complex Bayesian inference (Hu et al., 2023). A common drawback of these methods is either unreliable or costly uncertainty estimation. In contrast, HARBOR integrates external probabilistic models for a more rigorous and principled calibration (Feng et al., 2025; Hou et al., 2023). Its unique contribution is the orchestration of two Bayesian models: The first is a classic Naïve Bayes classifier (Zhang, 2004) whose primary limitation is its well-known conditional independence assumption (Prabha et al., 2022; He et al., 2025). To address this, we introduce a Latent-Augmented Causal Bayesian Network (CBN) that explicitly models dependencies between factors. Within this framework, the LLM is not only used for parameter elicitation (Nafar et al., 2025) but is also employed as a dynamic causal discovery engine to define the CBN's structure. By aggregating predictions from these models, HARBOR synthesizes their respective strengths to achieve a more reliable probability estimate.

**Structured Retrieval for Grounded Reasoning**   Recent work in Retrieval-Augmented Generation (RAG) (Lewis et al., 2020) has moved towards structured knowledge sources, employing graphs and hierarchies to address challenges like the "knowledge gap" between local and global context (Edge et al., 2024; Huang et al., 2025; Zhang et al., 2024). These studies collectively demonstrate the benefits of hierarchical and hybrid retrieval strategies. While these advanced methods primarily focus on indexing and structuring existing document corpora, HARBOR distinguishes itself by proactively constructing its knowledge source from the ground up. Our approach first generates a comprehensive set of reasoning factors, then organizes them into an interpretable hierarchy using established clustering techniques (Simeone et al., 2023) and LLM-guided theming (Zhang et al., 2023; Azher et al., 2024). This process yields a purpose-built structure tailored specifically for reasoning, enabling a sophisticated retrieval process founded on high-quality, contextually-aware evidence, rather than relying on a general-purpose indexed document store.

## 3   PRELIMINARIES

### 3.1   PROBLEM FORMULATION

We formulate reliable decision-making as a contextual binary inference task. Given a context $C$ and two competing hypotheses, $O_1$ and $O_2$, the objective is to estimate the conditional probabilities $P(O_i \mid C)$ for $i \in \{1, 2\}$. A calibrated estimate allows a system to robustly decide which hypothesis is more plausible. Following Feng et al. (2025), we decompose $C$ into a high-level scenario $S_{\text{cen}}$ and a downstream condition $u$, enabling a neutral reasoning space based on $S_{\text{cen}}$ that mitigates biases from $u$. For completeness, we treat an "unknown" prediction as abstention: letting $\mathcal{F}^\star$ denote the factor–mapping operator (see §4.2), the system yields an unknown prediction iff $|\mathcal{F}^\star(u)| = 0$ or $\max_{i \in \{1,2\}} P(O_i \mid C) < \tau$ for a preset $\tau \in (0, 1)$.

### 3.2   ABDUCTIVE-DEDUCTIVE INFERENCE FRAMEWORK

Some prior work (Feng et al., 2025) implements a two-stage process. First, in a step of *forward abduction*, it generates a set of $N$ discrete factors $\mathcal{F} = \{F_1, \ldots, F_N\}$ directly from the scenario $S_{\text{cen}}$. Each factor $F_j$ has a corresponding value set $\mathcal{V}_j$. The Cartesian product of these value sets, $\mathcal{V} = \mathcal{V}_1 \times \cdots \times \mathcal{V}_N$, forms the complete *information space*. An element $f = (f_1, \ldots, f_N) \in \mathcal{V}$ represents one fully-specified state of the world.

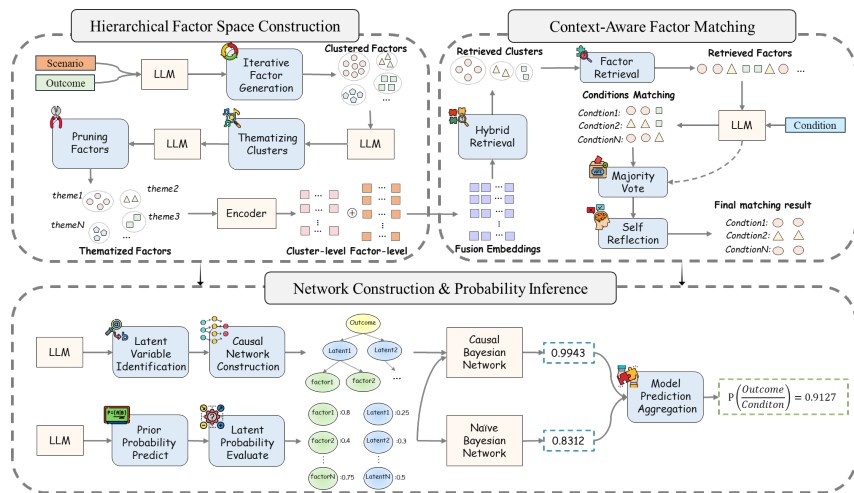

Figure 2: Overview of HARBOR: (1) *Factor–Space Construction*: iterative factor generation and hierarchical clustering generate a dense, two-level factor hierarchy; (2) *Context–Aware Mapping*: perform coarse-to-fine retrieval over the factor hierarchy, then apply self-consistent filtering and reflective refinement to select factors relevant to the condition; (3) *Inference Orchestration*: construct Naïve Bayes and Causal Bayesian networks from the mapped factors—using an LLM to identify latent variables and combine their outputs into a single calibrated probability.

Next, the framework *deductively* computes the outcome probability by marginalizing over this information space: $P(O_i \mid C) = \sum_{f \in \mathcal{V}} P(O_i \mid f) \, P(f \mid C)$.

Here, $P(O_i \mid f)$ is the conditional probability table (CPT) and $P(f \mid C)$ is the probability of instance $f$ given the context. To make this tractable, factors are assumed to be conditionally independent given $C$, simplifying the calculation to: $P(O_i \mid C) = \sum_{f \in \mathcal{V}} P(O_i \mid f) \prod_{j=1}^{N} P(f_j \mid C)$

## 4 METHODOLOGY

In this section, we introduce HARBOR, a three-stage framework designed to transform raw LLM outputs into well-calibrated probabilities. As illustrated in Figure 2, the process consists of factor-space construction, context-aware mapping, and probabilistic inference.

### 4.1 ITERATIVE ABDUCTION FOR FACTOR–SPACE EXPANSION

#### 4.1.1 BOTTOM-UP ABDUCTION STRATEGY

We define *bottom-up abduction* as a two-stage process that inverts the traditional paradigm by decoupling factor generation from structuring. First, we iteratively generate a comprehensive set of factors, and then we cluster and theme them to build a structured factor space.

The generation stage begins with an empty set $\mathcal{F}^{(0)} = \varnothing$ and iterates until reaching a target size $K$ or exceeding $T_{\max}$ rounds: (a) **Contextual sentence generation**, where a few-shot prompt elicits $b$ diverse supporting or refuting sentences per scenario, explicitly encouraging varied reasoning chains and broad aspect coverage; and (b) **Factor harvesting and validation**, where an LLM extraction prompt identifies all distinct factors in the sentences, which are parsed into a candidate set $\Delta\mathcal{F}$, validated for semantic non-redundancy and domain relevance, and merged into the main set via $\mathcal{F}^{(t+1)} = \mathcal{F}^{(t)} \cup \Delta\mathcal{F}$.Our factor-generation procedure achieves geometric convergence in recall completeness, as detailed in A.2.

#### 4.1.2 SELF-CONSISTENCY AND STRUCTURAL ORGANIZATION

Once the factor set $\mathcal{F}$ is sufficiently large, we impose a hierarchical structure through a four-stage pipeline: (i) encode each factor $f$ into $\mathbb{R}^d$ using MiniLM (Wang et al., 2020); (ii) apply UMAP to

project embeddings into a lower-dimensional space, reducing noise while preserving local semantic neighborhoods (McInnes et al., 2018); (iii) run HDBSCAN to discover cohesive factor groups without pre-specifying the cluster count (McInnes et al., 2017); and (iv) prompt an LLM to assign a concise theme (e.g., *Economic Feasibility*) to each cluster and remove redundant factors, yielding a final hierarchical structure $\widetilde{\mathcal{F}}$. To derive factor attributes—supporting $O_1$, supporting $O_2$, or neutral—for downstream tasks, we prompt the LLM to classify each factor as "supports $O_1$", "supports $O_2$", or "neutral".

## 4.2 CONTEXT-AWARE FACTOR MAPPING

With the hierarchical factor space $\widetilde{\mathcal{F}}$ in place, mapping a downstream condition $u$ via brute-force search is computationally intractable. To address this, we propose a multi-stage pipeline that defines a mapping operator $\mathcal{F}^\star$: starting from a broad retrieval stage to maximize recall, then applying targeted filtering and precision-driven refinement, it maps a condition $u$ to a compact, high-confidence factor set $\mathcal{F}^\star(u)$.

### 4.2.1 HIERARCHICAL RETRIEVAL FOR CANDIDATE GENERATION

Our retrieval design leverages the two-level structure of the factor space $\widetilde{\mathcal{F}}$ to efficiently generate a high-recall candidate set, $\mathcal{F}_{\text{cand}}(u)$. The process begins by embedding the condition $u$ and all factor cluster prototypes into a shared vector space $\mathbb{R}^d$. A key design element is the cluster prototype itself, formulated as a weighted average of its thematic label and its member factor embeddings:

$$e_{C_j} = \alpha \cdot e_{\text{theme}} + (1 - \alpha) \cdot \frac{1}{|F_j|} \sum_{f \in F_j} e_f \qquad (1)$$

where $e$ denotes an embedding vector and $\alpha \in [0, 1]$ is a weighting parameter. This formulation balances high-level categorical meaning with fine-grained distributional semantics.

We then perform a **coarse-to-fine search** using K-Nearest Neighbors (KNN) (Guo et al., 2003) to rapidly zero in on relevant factors:

1. **Coarse Search (Cluster-Level):** We first identify the top-$K_1$ clusters whose prototypes are most semantically similar to the condition embedding $e_u$. This step acts as a high-level filter, dramatically narrowing the search space to the most promising regions.

2. **Fine Search (Factor-Level):** Within this curated set of clusters, we then conduct a more granular search, retrieving the top-$K_2$ individual factors from each selected cluster that are closest to $e_u$.

The union of factors retrieved from this two-step process forms the candidate set $\mathcal{F}_{\text{cand}}(u)$, which is intentionally broad to ensure no relevant factors are prematurely discarded.

### 4.2.2 SELF-CONSISTENT FILTERING FOR ROBUST SELECTION

The high-recall nature of our retrieval strategy may introduce noise. To address this, we perform a robust filtering step to identify the most pertinent factors from the candidate set. Emulating the principles of self-consistency (Wang et al., 2022), we query an LLM $R$ times with the same prompt, asking it to select the subset of $\mathcal{F}_{\text{cand}}(u)$ that is directly supported by the condition $u$. We aggregate the results by tallying votes for each factor using the formula $v_f(u) = \sum_{r=1}^{R} \mathbf{1}[f \in m^{(r)}(u)]$, where $m^{(r)}(u)$ is the subset returned in the $r$-th query. We then retain only those factors whose vote count meets a configurable threshold $\gamma$ (e.g., $\gamma = \lceil \text{vote\_ratio} \cdot R \rceil$). This majority-vote mechanism yields a filtered set, $\mathcal{F}_{\text{vote}}(u)$, which effectively prunes spurious or weakly supported factors.

### 4.2.3 REFLECTIVE REFINEMENT FOR FINAL PRECISION

As a final quality control step, the filtered set $\mathcal{F}_{\text{vote}}(u)$ is submitted to a specialized reflection prompt. This prompt instructs the LLM to review and prune any remaining factors that, upon second inspection, lack clear and direct relevance to the condition $u$. This structured self-critique has been shown to reduce hallucinations and improve factual precision (Ji et al., 2023), yielding the final, high-precision factor set $\mathcal{F}^\star(u)$.

## 4.3 Probabilistic Inference with Elicited Parameters

The final stage of our framework transforms the mapped factor set, $\mathcal{F}^*(u)$, into a calibrated probability for the competing hypotheses, $O_1$ and $O_2$. To achieve this, we first define the structure of two probabilistic models—a Naïve Bayes (NB) model and a Causal Bayesian Network (CBN). We then describe the process of eliciting the necessary parameters from an LLM and, finally, how these parameters are used to perform inference in each model.

**Naïve Bayes Model.** The Naïve Bayes model is structured on the assumption that all factors are conditionally independent given the hypothesis. It consists of a root *Outcome* node (representing $O_1$ and $O_2$) with directed edges to each child factor node $f_j \in \mathcal{F}^*(u)$.

**Latent-Augmented Causal Bayesian Network.** To capture dependencies between factors, we construct a CBN whose structure is learned dynamically for each scenario. We prompt an LLM to act as a causal discovery engine: given the list of relevant factors, it identifies a set of latent variables $\mathcal{L} = \{L_1, \ldots, L_k\}$ and partitions the factors among them. In this model, the binary states of each latent variable, $L_i = 1$ and $L_i = 0$, serve as proxies for the final hypotheses $O_1$ and $O_2$ define the network's edges. Specifically, there is a directed edge from each latent variable to the factors it groups, represented as $L_i \to f_j$, and a directed edge from the Outcome node to each latent variable, represented as Outcome $\to L_i$. Outcomes serve as roots and the Latents as their shared child, relaxing NB's independence assumption by making factors independent only when conditioned on their latent parent.

### 4.3.1 Parameterization and Inference

Inspired by prior work validating the use of LLMs to elicit informative priors (Gouk & Gao, 2024; Thambawita & Hammer, 2024; Nafar et al., 2025), our framework efficiently parameterizes its Bayesian networks. We directly query the LLM to obtain the necessary conditional probabilities—both at the factor and latent levels—thus bypassing the need for costly data sampling.

**Factor-Level Parameters**: For each factor $f$, we elicit its posterior probability given hypothesis $O_1$, denoted as $\phi_f = P(O_1 \mid f)$. This captures the evidential strength of a single factor in favor of $O_1$. For the Naïve Bayes (NB) model, which is parameterized by likelihoods $P(f \mid O_k)$, we adopt a simple binary, symmetric-prior approximation and set $\theta_f = P(f \mid O_1) \approx \phi_f, P(f \mid O_2) \approx 1 - \phi_f$.
**Latent-Level Parameters**: Exclusively for the Causal Bayesian Network (CBN), we elicit the conditional probability of each latent variable $L_i$ given a hypothesis, such as $P(L_i = 1 \mid O_1)$ and $P(L_i = 0 \mid O_2)$. To ensure numerical stability, all elicited probabilities are smoothed. These parameters are then used to build and perform inference in our two probabilistic models.
The NB model is parameterized by using the elicited posterior $\phi_f$ as a proxy for the likelihood, defining the CPT for each factor $f$ as $P(f \mid O_1) = \theta_f \approx \phi_f, P(f \mid O_2) \approx 1 - \phi_f$. Given an evidence set $\mathcal{E}$ and assuming a uniform prior over outcomes, the posterior probability for hypothesis $O_1$ is calculated as:

$$P(O_1 \mid \mathcal{E}) = \frac{\prod_{f \in \mathcal{E}} P(f \mid O_1)}{\prod_{f \in \mathcal{E}} P(f \mid O_1) + \prod_{f \in \mathcal{E}} P(f \mid O_2)}. \tag{2}$$

The CBN's CPTs are parameterized as follows. **Latent Nodes** $(L_i)$, as root nodes, are given a uniform prior, so $P(L_i = 1) = 0.5$. The CPT for a **Factor Node** $(f_j)$, conditioned on its latent parent $L_i$, uses the same NB-style likelihood parameters, specifically $P(f_j \mid L_i = 1) = \theta_{f_j} \approx \phi_{f_j}, P(f_j \mid L_i = 0) \approx 1 - \phi_{f_j}$. The CPT for the **Outcome Node**, $P(\text{Outcome} \mid L_1, \ldots, L_k)$, is derived from the elicited latent-level parameters ($P(L_i \mid O_k)$) via Bayes' rule.

Given the evidence $\mathcal{E}$, we perform exact inference using the variable elimination algorithm (Zhang & Poole, 1994) to compute the posterior $P(\text{Outcome} = O_1 \mid \mathcal{E})$.

### 4.3.2 Model Aggregation

To synthesize the predictions from our NB and CBN models into a single, more reliable estimate, we employ the Linear Opinion Pool (LOP) aggregation strategies: The LOP is a simple and powerful method that forms a weighted average of individual model predictions (Stone, 1961; Neyman &

Roughgarden, 2023; Stratigakos et al., 2024). We select it as our primary strategy for its computational efficiency and proven robustness. The aggregated probability is a convex combination: $P_{\text{LOP}}(O_1 \mid \mathcal{E}) = \sum_{M \in \{\text{NB,CBN}\}} w_M \cdot P(O_1 \mid \mathcal{E}, M)$, where the weights $w_M$ are fixed and sum to 1.

## 5 EXPERIMENT

### 5.1 EXPERIMENTAL SETUPS

**Datasets**   We evaluate our model on three reasoning and planning benchmarks from Feng et al. (2025), each formatted as a decision-support tuple $(S_{\text{cen}}, U, O)$: **Today** (1,000 instances), **Plasma** (279 scenarios, 1,395 instances) and **Common2Sense** (216 scenarios, 3,822 instances), the Common2Sense's test split being manually expanded under an unbiased protocol to boost diversity. To assess decision-making beyond planning, we further sample 100 examples from each of the four fact-checking datasets—**XSum** (Tang et al., 2024), **CNN** (Tang et al., 2024), **ExpertQA** (Malaviya et al., 2023) and **COVID** (Saakyan et al., 2021)—treating each document as condition $U$ and each claim as outcome $O$, with $S_{\text{cen}}$ left empty. (dataset details refer to A.5).

**Settings**   We conduct all experiments with four LLMs: Qwen2.5-32b, Qwen2.5-72b (Bai et al., 2023), DeepSeekV3-671b (Liu et al., 2024), and GPT-4 (Achiam et al., 2023). The Qwen models were deployed locally, while DeepSeekV3-671b was accessed through DeepSeek's official API. We set $K_1 = 3$ and $K_2 = 5$ in the KNN search; more hyperparameters and settings are provided in Appendix A.6. All experiments ran on 4× NVIDIA RTX 4090 (24GB) GPUs.

**Preference-based Pairwise Evaluation**   To begin, we evaluate our model on Common2Sense using the *preference-based pairwise evaluation* framework introduced by Feng et al. (2025). The experimental setup is as follows. Given a scenario $S_{\text{cen}}$ and two potential outcomes, $O_1$ and $O_2$, we select two distinct additional conditions, $U_1$ and $U_2$. Crucially, both conditions are constructed to support the same outcome, $O_1$, over $O_2$. These are then used to form two slightly different contexts: $C_1 = S_{\text{cen}} + U_1$ and $C_2 = S_{\text{cen}} + U_2$. The model's task is to assign conditional probabilities $P(O_i|C_1)$ and $P(O_i|C_2)$ for each outcome $i \in \{1, 2\}$. A successful evaluation requires the model to correctly discern the subtle difference in the degree of support between the two contexts. For instance, if human judgment determines that $C_1$ provides stronger support for $O_1$ than $C_2$ does, we expect the model's probability assignments to satisfy the following relationship: $P(O_1|C_1) > P(O_1|C_2) > P(O_2|C_2) > P(O_2|C_1)$, we set the abstention threshold $\tau = 0$, so the system yields an unknown prediction only when no factors are matched, i.e., $|\mathcal{F}^\star(u)| = 0$.

**Decision Making Evaluation**   We evaluate HARBOR's decision-making on reasoning and planning datasets (Plasma, Today) and four fact-checking datasets for generalization (ExpertQA, XSum, COVID, CNN). In each task, given a context $C$ and two outcomes $O_1$ and $O_2$, the model must select the more plausible one. A decision is considered correct if the chosen outcome has a higher conditional probability, e.g., satisfying $P(O_1|C) > P(O_2|C)$ when $O_1$ is the ground truth.

**Baseline Methods**   We compare our approach against several baselines for probability estimation from LLMs in our experiments: (1) **BIRD**: Builds a structured factor space via single-pass abductive reasoning and uses a Naïve Bayes model for inference (Feng et al., 2025). (2) **Vanilla**: Prompts the model to verbalize its estimated probability (Wang et al., 2022). (3) **Logits**: Converts the normalized token probability of the decision token into a probability score. (4) **CoT**: Elicits a chain-of-thought reasoning process before asking for the final probability (Wei et al., 2022). (5) **Compare**: Selects which of two conditions better supports the outcome without producing individual probabilities, an unfair setting for methods relying on those estimates (Feng et al., 2025). (6) **Factor-based**: Averages probabilities over five factors generated with knowledge of the gold outcome, making direct comparison invalid.

### 5.2 RESULTS

The results are listed in Table 1. Our framework, HARBOR, exhibits a superior alignment with human preference, achieving a top F1 of 59.6%, more than 14% higher than DeepSeek-V3-671b. This demonstrates that HARBOR produces reliable probability estimations.

| Method | Model | Context1 | Context2 | Same | Average | Coverage |
|---|---|---|---|---|---|---|
| Random | Guess | 0.333 | 0.333 | 0.333 | 0.333 | – |
| CoT | Qwen2.5-32b | 0.394 | 0.370 | 0.118 | 0.319 | – |
| | Qwen2.5-72b | 0.370 | 0.382 | 0.159 | 0.311 | – |
| | DeepSeek-V3-671b | 0.414 | 0.395 | 0.146 | 0.338 | – |
| Vanilla | Qwen2.5-32b | 0.528 | 0.528 | 0.211 | 0.462 | – |
| | Qwen2.5-72b | 0.526 | 0.528 | 0.183 | 0.470 | – |
| | DeepSeek-V3-671b | 0.489 | 0.498 | 0.137 | 0.425 | – |
| Logits | Qwen2.5-32b | 0.504 | 0.491 | 0.150 | 0.445 | – |
| | Qwen2.5-72b | 0.523 | 0.500 | 0.156 | 0.447 | – |
| | DeepSeek-V3-671b | 0.516 | 0.513 | 0.162 | 0.453 | – |
| Compare | Qwen2.5-32b | 0.583 | 0.480 | 0.221 | 0.477 | – |
| | Qwen2.5-72b | 0.566 | 0.570 | 0.267 | 0.534 | – |
| | DeepSeek-V3-671b | 0.560 | 0.578 | 0.286 | 0.542 | – |
| | GPT-4 | 0.587 | 0.548 | 0.302 | 0.556 | – |
| Factor-based | Qwen2.5-32b | 0.556 | 0.564 | 0.054 | 0.525 | – |
| | Qwen2.5-72b | 0.509 | 0.509 | 0.250 | 0.496 | – |
| | DeepSeek-V3-671b | 0.553 | 0.546 | 0.165 | 0.506 | – |
| BIRD | Qwen2.5-72b[†] | 0.497 | 0.412 | 0.271 | 0.421 | 87.77% |
| | Qwen2.5-72b[‡] | 0.567 | 0.520 | **0.307** | 0.513 | 99.99% |
| HARBOR (ours) | Qwen2.5-32b | 0.612 | 0.587 | 0.152 | 0.572 | 99.79% |
| | Qwen2.5-72b | **0.626** | **0.627** | 0.281 | **0.596** | 99.95% |

Table 1: Preference-based pairwise evaluation using F1 score on the Common2sense dataset. Context1 is the F1 score for the class where the first context better supports the outcome than the second, and Context2 is for the reverse. The same is the F1 score for cases where both contexts offer equal support. Avg is the global micro-averaged F1 score. BIRD[†] uses the original, unmodified mapping output of the BIRD method. BIRD[‡] employs a relaxed mapping condition, forcing the model to match each condition with at least one factor from the factor space. Unlike BIRD, which manually filtered "unknown" cases, we remove this intervention for fairness.

Some baselines, such as Compare, perform joint context processing, giving them an unfair advantage. We also introduce Factor-based, another unfair baseline that assumes the model can construct a perfect factor space and always match the gold-aligned factor. Despite these advantages, HARBOR still outperforms both baselines—achieving up to 5% in F1. Even our smaller Qwen2.5-32b model outperforms the baselines built on the larger Qwen2.5-72b. In terms of coverage, HARBOR achieves near-complete coverage with its default mapping—no manual intervention required—while BIRD attains only 87.8% under its original mapping and must relax matching to reach comparable coverage.

HARBOR is designed to be highly sensitive to nuanced differences between contexts, which results in its F1 being lower on the Same class compared to other categories. Given the severe class imbalance (the Same class is only 9.82% of the dataset), the Micro Avg F1 score provides a more meaningful measure of overall performance, where our method clearly excels.

For decision-making, following Feng et al. (2025), when the BIRD baseline predicts "unknown", we default to the CoT method to ensure a decision is made. As other methods do not predict "unknown", we omit the coverage metric. ExpertQA, XSum, CNN, and Today represent more challenging settings due to long contexts

| Model | expertqa | xsum | covid | cnn | today | plasma |
|---|---|---|---|---|---|---|
| *DeepSeek-V3-671b* | | | | | | |
| Vanilla | 0.538 | 0.537 | 0.752 | 0.495 | 0.710 | 0.774 |
| Vanilla * | 0.562 | 0.483 | 0.686 | 0.500 | 0.647 | 0.781 |
| CoT | 0.541 | 0.527 | 0.685 | 0.495 | 0.783 | 0.784 |
| CoT * | 0.498 | 0.517 | **0.754** | 0.462 | 0.803 | **0.850** |
| *Qwen2.5-72b* | | | | | | |
| Vanilla | 0.517 | 0.557 | 0.538 | 0.604 | 0.583 | 0.604 |
| Vanilla * | 0.530 | 0.473 | 0.686 | 0.500 | 0.597 | 0.649 |
| CoT | 0.590 | 0.531 | 0.721 | 0.589 | 0.593 | 0.565 |
| CoT * | 0.552 | 0.484 | 0.701 | 0.500 | 0.713 | 0.745 |
| BIRD | 0.585 | 0.495 | 0.670 | 0.528 | 0.741 | 0.724 |
| HARBOR(ours) | **0.605** | **0.567** | 0.730 | **0.629** | **0.827** | 0.762 |

Table 2: Balanced accuracy on fact-checking tasks and accuracy on reasoning/planning tasks. Methods marked * make direct decisions without probability outputs.

or temporal reasoning, while COVID and Plasma are relatively simpler (Feng et al., 2025). For decision-making on fact-checking datasets, we convert probabilistic outputs into support labels by applying a fixed threshold, following prior work in fact verification (Jayaweera et al., 2024). For the Plasma and Today datasets, we adhere to the evaluation of (Feng et al., 2025). As shown in Table 2, HARBOR consistently outperforms all baselines at the 72b scale and even surpasses the 671b model on the harder tasks.

## 5.3 ABLATION STUDY

To validate our design, we conducted an ablation study on Common2Sense (Table 3), analyzing five key variants. In addition to ablating core components, the study compares our standard LOP with an alternative aggregation strategy: Bayesian Model Averaging (BMA), a theoretically rigorous method that weights models by their posterior probabilities (Hoeting et al., 1999; Fragoso et al., 2018).

The five variants are: (1) **w/o cbn**, which uses only the Naïve Bayes model in the aggregation step; (2) **w/o nb**, which uses only the Causal Bayesian Network model in the aggregation step; (3) **w/o cluster**, which expands factors without hierarchical clustering, equivalent to simply increasing the number of factors; (4) **w/o hierarchy**, which retains clustering but removes weighted fusion and coarse-to-fine retrieval; (5) **w/o pe-llm**, which replaces LLM-elicited parameters with frequency-based estimates (A.3); and (6) **HARBOR (BMA)**, which implements the BMA strategy (A.4).

| Component | Qwen2.5-32b | | | | | Qwen2.5-72b | | | | |
|---|---|---|---|---|---|---|---|---|---|---|
| | Ctx. 1 | Ctx. 2 | Same | Avg. | Cov. | Ctx. 1 | Ctx. 2 | Same | Avg. | Cov. |
| w/o cbn | 0.592 | 0.579 | 0.152 | $0.558_{\downarrow0.014}$ | – | 0.530 | 0.518 | 0.219 | $0.506_{\downarrow0.090}$ | – |
| w/o nb | 0.518 | 0.514 | 0.197 | $0.479_{\downarrow0.093}$ | – | 0.592 | 0.583 | 0.260 | $0.557_{\downarrow0.039}$ | – |
| w/o cluster | 0.480 | 0.497 | 0.193 | $0.457_{\downarrow0.115}$ | 98.93% | 0.420 | 0.429 | 0.173 | $0.400_{\downarrow0.196}$ | 98.82% |
| w/o hierarchy | 0.514 | 0.511 | 0.131 | $0.491_{\downarrow0.081}$ | 99.84% | 0.515 | 0.512 | 0.127 | $0.491_{\downarrow0.105}$ | 99.92% |
| w/o pe-llm | 0.459 | 0.484 | 0.271 | $0.435_{\downarrow0.137}$ | – | 0.505 | 0.549 | 0.281 | $0.483_{\downarrow0.113}$ | – |
| HARBOR (BMA) | 0.615 | 0.592 | 0.152 | $0.575_{\uparrow0.003}$ | – | 0.604 | 0.621 | 0.281 | $0.592_{\downarrow0.004}$ | – |

Table 3: Ablation study results based on Table 1. Ctx. 1/2, Same, Avg. and Cov. denote F1 for Context 1, Context 2, Same classes, micro-averaged F1 and coverage

The "–" in the Coverage column indicates that the variant did not affect coverage. While coverage remains near 100% for most variants, the dramatic drop in (2) confirms that unstructured factor expansion is insufficient. The performance declines in (4) and (3) underscore the importance of accurate parameter elicitation and our two-level hierarchy, respectively. Finally, both aggregation strategies—LOP and BMA—achieve similarly high coverage and comparable F1 scores, demonstrating that effective fusion, combined with hierarchical mapping and precise elicitation, is key to HARBOR's robust performance.

On the Common2sense dataset with Qwen2.5-72B, we plug the original BIRD[‡] pipeline into HARBOR's factor space while keeping BIRD-style CPT training. Within this unified setup, we compare four variants: the BIRD[‡] baseline with HARBOR's generated factors, HARBOR's factors and mapping, and the latter further augmented with a CBN combined with Naïve Bayes via a linear opinion pool. Table 4 reports Avg (global micro-averaged F1 over CONTEXT1, CONTEXT2, and SAME, as in Table 1), coverage, and mapping-stage cost normalized by the BIRD[‡] baseline (Tok./B, Time/B). Detailed training and sampling procedures are given in Appendix A.10.

As shown in Table 4, replacing BIRD[‡]'s original factors with HARBOR's denser factor space improves Avg from 0.513 to 0.532 but increases mapping cost by almost 3×. Using HARBOR's context-aware factor mapping on the same factor space recovers efficiency while still outperforming the BIRD[‡] baseline. Adding CBN aggregation on top of HARBOR's factors and mapping yields the strongest variant at essentially the same reduced cost, indicating that factor-space construction, mapping, and CBN aggregation all contribute beyond a stronger factor generator alone.

| Setting | Avg | Cov. | Tok./B | Time/B |
|---|---|---|---|---|
| BIRD[‡] baseline | 0.513 | 99.99% | 1.00 | 1.00 |
| +F | 0.532 | 99.12% | 2.98 | 2.87 |
| +F+M | 0.526 | 98.81% | 0.17 | 0.55 |
| +F+M+CBN | **0.568** | 98.81% | 0.17 | 0.55 |

Table 4: BIRD[‡] variants on HARBOR's factor space and mapping on Common2sense.

## 5.4 ANALYSIS

Figure 3 reports the computation time, token usage, and API calls on the Common2Sense dataset, using Qwen2.5-72B under the same experimental setup as the main results in Table 1. On average, per scenario, HARBOR runs in about 0.74× the time, uses 0.24× the tokens, and makes 0.37× the API calls of BIRD on this benchmark. While HARBOR invests more in factor-space construction,

its hierarchical condition–factor mapping significantly reduces downstream cost. For HARBOR, as more factors are added, the unknown rate falls and the average F1 generally improves, reflecting a trade-off between coverage and accuracy. More detailed analyses are provided in Appendix A.7 (e.g., retrieval algorithms, the effects of $K_1$ and $K_2$ and analyses of the probability-estimator LLM).

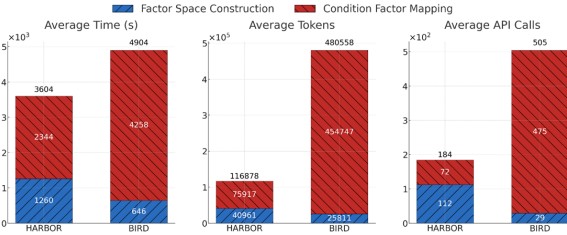

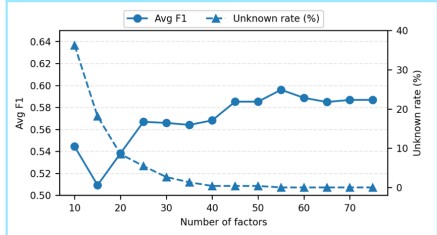

(a) Cost analysis of HARBOR vs. BIRD: average per-scenario runtime, token usage, and API calls by stage.

(b) Unknown rate and F1 scores for HARBOR as the number of factors increases.

Figure 3: Cost and coverage–accuracy analysis for HARBOR and BIRD.

### 5.4.1 UNCERTAINTY ESTIMATES AND STATISTICAL ANALYSIS

To quantify uncertainty in our main results, we run both HARBOR and BIRD five times under the same settings as Tables 1 and 2, and report the mean, standard deviation, and 95% confidence intervals using a $t$-distribution (Student, 1908). Table 5 summarizes F1 for the preference-based evaluation, and Table 6 reports (balanced) accuracy on the decision-making benchmarks. Across all preference metrics, HARBOR consistently outperforms BIRD with gains larger than the corresponding standard deviations; on the decision-making datasets, HARBOR with Qwen2.5-72B likewise achieves uniformly higher accuracy with small variance. The single-run scores in Tables 1 and 2 all fall within the 95% confidence intervals in Tables 5 and 6, indicating that those point estimates are representative rather than outliers.

| Model | Metric | Context1 | Context2 | Same | Average |
|---|---|---|---|---|---|
| BIRD · Qwen-72B | Mean $\pm$ Std | $0.568 \pm 0.012$ | $0.523 \pm 0.014$ | $0.306 \pm 0.011$ | $0.526 \pm 0.013$ |
| | 95% CI | [0.553, 0.584] | [0.505, 0.540] | [0.293, 0.319] | [0.511, 0.542] |
| HARBOR · Qwen-32B | Mean $\pm$ Std | $0.602 \pm 0.012$ | $0.583 \pm 0.012$ | $0.146 \pm 0.018$ | $0.563 \pm 0.017$ |
| | 95% CI | [0.587, 0.617] | [0.569, 0.598] | [0.124, 0.169] | [0.542, 0.583] |
| HARBOR · Qwen-72B | Mean $\pm$ Std | $0.610 \pm 0.021$ | $0.611 \pm 0.035$ | $0.278 \pm 0.014$ | $0.606 \pm 0.030$ |
| | 95% CI | [0.584, 0.637] | [0.567, 0.655] | [0.260, 0.295] | [0.570, 0.643] |

Table 5: Preference-based evaluation (F1) with mean $\pm$ standard deviation and 95% confidence intervals over 5 runs.

| Model | Metric | expertqa | xsum | covid | cnn | today | plasma |
|---|---|---|---|---|---|---|---|
| HARBOR · Qwen-72B | Mean $\pm$ Std | $0.611 \pm 0.019$ | $0.564 \pm 0.016$ | $0.726 \pm 0.013$ | $0.621 \pm 0.029$ | $0.817 \pm 0.031$ | $0.764 \pm 0.022$ |
| | 95% CI | [0.587, 0.634] | [0.544, 0.583] | [0.710, 0.742] | [0.585, 0.657] | [0.778, 0.856] | [0.736, 0.791] |
| BIRD · Qwen-72B | Mean $\pm$ Std | $0.582 \pm 0.023$ | $0.511 \pm 0.022$ | $0.664 \pm 0.020$ | $0.540 \pm 0.027$ | $0.754 \pm 0.035$ | $0.718 \pm 0.022$ |
| | 95% CI | [0.552, 0.611] | [0.484, 0.539] | [0.639, 0.689] | [0.507, 0.574] | [0.711, 0.797] | [0.691, 0.746] |

Table 6: Decision-making accuracy / balanced accuracy with mean $\pm$ standard deviation and 95% confidence intervals over 5 runs.

## 6 CONCLUSION

We introduced the HARBOR framework to generate reliable and calibrated probability estimates from Large Language Models for critical decision-making. By integrating a hierarchical factor-space construction with the orchestration of Naïve Bayes and a Causal Bayesian Network, our framework showed notable enhancements in decision-making accuracy and alignment with human judgment. Our experimental results underscore the effectiveness of HARBOR, thereby increasing the practical utility of LLMs in high-stakes scenarios and fostering more trustworthy autonomous systems.

ETHICS STATEMENT

This work aims to improve reliable probability estimation and interpretable decision-making under incomplete information, with clear social value in domains such as emergency response and public governance. HARBOR reduces the risks of "unknown" predictions and overconfidence through hierarchical abduction and Bayesian orchestration, while providing an auditable factor space as the basis of decisions. However, HARBOR measures reasoning consistency and uncertainty calibration, which is not equivalent to factual correctness or fairness; if training data or prompts contain bias, the model may still produce highly confident yet biased conclusions. Therefore, HARBOR should be used in conjunction with fact verification, bias detection, and human oversight, particularly in high-stakes or sensitive scenarios. No personal or sensitive data are used or released in this study, and all resources comply with platform and licensing requirements.

REPRODUCIBILITY STATEMENT

To ensure reproducibility, we provide methodological details and experimental configurations, with additional resources in the appendix:

- **Algorithmic details:** Section 4 describes the three-stage pipeline (factor-space construction, hierarchical retrieval and mapping, probabilistic inference), with structured pseudocode in Appendix A.9.

- **Hyperparameters:** All key hyperparameters (e.g., $K_1{=}3$, $K_2{=}5$, $\alpha$, $R$, $\epsilon$) are listed in Table 8 (Appendix A.6), together with their values and rationale.

- **Datasets and preprocessing:** Data sources and construction are described in Section 5.1 and Appendix A.5, with dataset statistics summarized in Table 7.

- **Experimental settings:** Model configurations are detailed in the Section 5.1.

- **Evaluation protocol:** Pairwise preference and decision evaluations are defined in Section 5.1, with main and ablation results presented in Tables 1 and 3.

- **Implementation and resources:** Key components for retrieval, clustering, and inference are detailed in Appendices A.8 and A.9. All source code and scripts required to reproduce our experiments are included in the supplementary material. In addition, prompt examples are provided in Appendix B.3, comparative examples between HARBOR and BIRD are given in Appendix B.2, and a statement of LLM usage in this work is presented in Appendix B.1.

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

# A   APPENDIX A

## A.1   UNCERTAINTY DECOMPOSITION

To understand how outcome probabilities are inferred from partial contextual information, we start with the law of total probability over the full factor space $\mathcal{F}$:

$$P(O_i \mid C) = \sum_{f \in \mathcal{F}} P(O_i \mid f, C) \cdot P(f \mid C) \tag{3}$$

We can divide the factor set into two disjoint parts: those consistent with context $C$, and those that contradict it. Denote these as:

$$\begin{aligned} \mathcal{F}_C &= \{\, f \in \mathcal{F} \mid f \cap C = f \,\}, \\ \mathcal{F}_{\neg C} &= \mathcal{F} \setminus \mathcal{F}_C. \end{aligned} \tag{4}$$

Then the probability becomes:

$$\begin{aligned} P(O_i \mid C) &= \sum_{f \in \mathcal{F}_C} P(O_i \mid f, C)\, P(f \mid C) + \sum_{f \in \mathcal{F}_{\neg C}} P(O_i \mid f, C)\, P(f \mid C) \\ &= \sum_{f \in \mathcal{F}_C} P(O_i \mid f)\, P(f \mid C) + \sum_{f \in \mathcal{F}_{\neg C}} P(O_i \mid f, C) \cdot 0 \\ &= \sum_{f \in \mathcal{F}_C} P(O_i \mid f)\, P(f \mid C). \end{aligned} \tag{5}$$

The second line follows from the assumption that only context-consistent factors have non-zero posterior $P(f \mid C)$, and the third line uses the independence assumption $P(O_i \mid f, C) = P(O_i \mid f)$ when $f \subseteq C$.

**Interpretation.** This formulation shows that outcome probability is a marginalization over consistent factors, where each term combines a factor's evidential support $P(O_i \mid f)$ with its relevance weight $P(f \mid C)$. Our retrieval and voting pipeline can thus be interpreted as approximating this sum with the selected set $\mathcal{F}^*(C)$:

$$P(O_i \mid C) \approx \sum_{f \in \mathcal{F}^*(C)} P(O_i \mid f)\, P(f \mid C) \tag{6}$$

## A.2   ASSUMPTIONS FOR PROBABILITY ESTIMATION IN HARBOR

**Weak ordering of factor odds ratios**   For each factor $f_j \in \mathcal{F}$, define its odds

$$o_j \;=\; \frac{P(O_1 \mid f_j)}{1 - P(O_1 \mid f_j)}, \tag{7}$$

We write

$$(o_1, \ldots, o_N) \;\succeq_E\; (o'_1, \ldots, o'_N). \tag{8}$$

if a decision maker is at least as confident in outcome $O_1$ given odds $(o_1, \ldots, o_N)$ as given $(o'_1, \ldots, o'_N)$. This relation is a weak ordering, satisfying:

(a) *Completeness*: for any two odds-vectors $A, B$, either $A \succeq_E B$ or $B \succeq_E A$;

(b) *Transitivity*: if $A \succeq_E B$ and $B \succeq_E C$, then $A \succeq_E C$.

**Non-interaction of factors** For any disjoint factor subsets $A, B \subseteq \mathcal{F}$, the ordering induced by $A$ alone is unchanged by the presence of $B$:

$$(o_j : f_j \in A) \succeq_E (o'_j : f_j \in A) \iff (o_j : f_j \in A \cup B) \succeq_E (o'_j : f_j \in A \cup B). \qquad (9)$$

**Self-consistency convergence** Querying the LLM $m$ times and taking a majority vote yields an error probability bounded by $\exp(-2m(q - 0.5)^2)$, where $q > 0.5$ is the accuracy of a single vote (Wang et al., 2022).

**Abduction recall guarantee** If each true factor is recovered in a given round with probability $p > 0$, then after $T$ rounds the probability of missing it is $(1 - p)^T$. Thus $T = O(\log(1/\varepsilon)/p)$ rounds suffice to cover the factor space with high probability (Cormode et al., 2010).

Together, these assumptions ensure that (1) factor odds admit a well-behaved weak ordering; (2) factors do not spuriously interact in the odds ranking; and (3) both abduction and self-consistency stages have explicit convergence guarantees.

## A.3 PARAMETER INITIALIZATION

**Factor prior initialization** Factor labels ("supports $O_1$", "supports $O_2$", "neutral") are assigned via the self-consistency procedure described in our method-Iterative Abduction. We then replace the LLM-elicited posterior $\theta_{f_j} = P(O_1 \mid f_j)$ with

$$P_{\text{init}}(O_1 \mid f_j) = \begin{cases} 0.75, & f_j \text{ labeled "supports } O_1\text{"}, \\ 0.50, & f_j \text{ labeled "neutral"}, \\ 0.25, & f_j \text{ labeled "supports } O_2\text{"}. \end{cases}$$

**Latent CPT computation** Each latent variable $L_k$ aggregates a subset of factors whose labels yield counts

$$\texttt{counts}[L_k] = \{\text{Outcome1} : c_k^1, \quad \text{Outcome2} : c_k^2, \quad \text{Neutral} : c_k^N\}.$$

We apply Laplace smoothing ($\alpha$) and split neutrals evenly:

$$\tilde{c}_k^1 = c_k^1 + \tfrac{1}{2} c_k^N + \alpha, \quad \tilde{c}_k^2 = c_k^2 + \tfrac{1}{2} c_k^N + \alpha,$$

then compute

$$P(L_k = 1 \mid O_1) = \frac{\tilde{c}_k^1}{\tilde{c}_k^1 + \tilde{c}_k^2}, \quad P(L_k = 1 \mid O_2) = \frac{\tilde{c}_k^2}{\tilde{c}_k^1 + \tilde{c}_k^2}.$$

Enumerating all $2^n$ latent-state vectors $x \in \{0, 1\}^n$, we form the likelihoods $\prod_k P(L_k = x_k \mid O_i)$, multiply by the prior $P(O_i) = 0.5$, and normalize:

$$P(x \mid O_i) = \frac{P(O_i) \prod_k P(L_k = x_k \mid O_i)}{\sum_{j=1}^{2} P(O_j) \prod_k P(L_k = x_k \mid O_j)}.$$

**Probability product approximation** We approximate the marginal likelihood of evidence $\mathcal{E}$ under model $M$ by

$$P(\mathcal{E} \mid M) \approx P(O_1 \mid \mathcal{E}, M) \cdot P(O_2 \mid \mathcal{E}, M).$$

## A.4 AGGREGATION METHODS

**Linear Opinion Pool (LOP)** Given per-model posteriors $P(O_1 \mid \mathcal{E}, M)$, LOP aggregates them as

$$P_{\text{LOP}}(O_1 \mid \mathcal{E}) = \sum_{M \in \{\text{NB}, \text{CBN}\}} w_M P(O_1 \mid \mathcal{E}, M).$$

We use fixed weights tailored to the underlying LLM: for Qwen2.5-32b, $w_{\text{NB}} = 0.8$ and $w_{\text{CBN}} = 0.2$; for Qwen2.5-72b, $w_{\text{NB}} = w_{\text{CBN}} = 0.5$.

**Bayesian Model Averaging (BMA)**    Under BMA, the final posterior is a weighted sum of each model's predictions:

$$P_{\text{BMA}}(O_1 \mid \mathcal{E}) = \sum_{M \in \{\text{NB,CBN}\}} P(M \mid \mathcal{E}) \, P(O_1 \mid \mathcal{E}, M),$$

where the model weights are proportional to the prior times the model evidence:

$$P(M \mid \mathcal{E}) \; \propto \; P(M) \, P(\mathcal{E} \mid M), \quad P(M) = \tfrac{1}{2}.$$

In practice, we approximate the evidence by the product of the two outcome posteriors, $P(\mathcal{E} \mid M) \approx P(O_1 \mid \mathcal{E}, M) \cdot P(O_2 \mid \mathcal{E}, M)$, and normalize these to obtain $P(M \mid \mathcal{E})$. Finally, the aggregated probability is

$$P_{\text{BMA}}(O_1 \mid \mathcal{E}) = \sum_M w_M \, P(O_1 \mid \mathcal{E}, M).$$

This "product-of-posteriors" is only an approximation: the true evidence requires summing over all latent or factor assignments, $P(\mathcal{E} \mid M) = \sum_x P(\mathcal{E}, x \mid M)$, which becomes intractable as the factor space grows.

## A.5    EXPERIMENT DATASET DESCRIPTION

**COMMON2SENSE**   Singh et al. (2021) introduces a multi-domain commonsense reasoning and planning benchmark comprising true/false natural language statements. We select only the *comparative reasoning* instances in which a smaller pre-trained model shows low confidence. Each statement is paired with its GPT-4–generated opposite, yielding two outcomes per scenario. For each outcome, 10 supporting conditions are generated and then filtered via reverse verification to ensure quality. The original test split contained 350 instances; to enable a fairer evaluation, the first three authors of this paper manually expanded it to 530 instances—strictly by adding contrasting conditions to the existing scenarios and outcomes, following the annotation protocol of Feng et al. (2025). The final dataset comprises 216 scenarios and 3,822 instances, with an average of 9 conditions per outcome.

**TODAY**   Feng et al. (2022) proposes a temporal reasoning dataset where the effect of appending an extra sentence on temporal relations is evaluated. It consists of 1,000 instances designed to probe fine-grained temporal inference under controlled scenario modifications.

**PLASMA**   Brahman et al. (2023) focuses on plan revision under a new condition. For each of 279 scenarios, GPT-4 identifies and rewrites the altered step into two alternative outcomes. The less common outcome in each pair is selected, and 5 supporting conditions are generated to favor it. The resulting dataset comprises 279 scenarios and 1,395 instances, with exactly 5 conditions per outcome.

**Fact-Checking Benchmarks**   In addition, we sample 100 instances (with random seed fixed to 42) from the test split of each of four widely used fact-checking datasets, treating the document as the condition $U$ and the claim as the outcome $O$, with the central scenario $S_{\text{cen}}$ left empty. The long-document contexts pose a challenge for robust factor-space construction.

- **COVID** (Saakyan et al., 2021): Scientific claims about COVID-19 paired with biomedical abstracts, reflecting high-precision domain reporting.

- **ExpertQA** (Malaviya et al., 2023): Expert-driven statements with detailed contextual explanations spanning medicine, law, and engineering.

- **CNN** (Tang et al., 2024): News-derived claims with multi-sentence evidential contexts, suitable for long-range factual consistency evaluation.

- **XSum** (Tang et al., 2024): Headline-style summaries paired with full article text, emphasizing abstractiveness under concise outcomes.

Table 7 reports the average character lengths of outcome and condition in each dataset.

| Dataset | Outcome | Condition |
|---------|---------|-----------|
| Common2Sense | 79.78 | 80.98 |
| Plasma | 92.69 | 79.76 |
| Today | 79.08 | 71.07 |
| COVID | 89.37 | 505.04 |
| ExpertQA | 163.20 | 2399.16 |
| CNN | 296.31 | 2972.89 |
| XSum | 127.25 | 1875.37 |

Table 7: Average character lengths of outcome and condition across all datasets.

## A.6 HYPERPARAMETER SETTINGS

Table 8 provides a comprehensive list of the key hyperparameters used in our experiments. **Decision Stage.** For downstream fact-checking, we convert probabilistic outputs into categorical labels by applying a fixed decision threshold: a claim is marked as support only when its predicted probability exceeds 0.9, otherwise it is treated as unsupport. Similar threshold-based rules are routinely adopted in recent fact-verification systems to balance precision and recall; for instance, AMREx employs dataset-specific entailment thresholds to separate "Supports", "Refutes", and "Not Enough Info" classes (Jayaweera et al., 2024).

| Stage | Parameter | Value | Description |
|-------|-----------|-------|-------------|
| **Iterative Factor Generation** | | | |
| Factor Generation | *For Common2Sense & Plasma datasets:* | | |
| | Target factor count ($K$) | 80 | Target number of unique factors. |
| | Batch size ($b$) | 10 | Sentences generated per LLM call. |
| | Max rounds ($T_{\max}$) | 20 | Maximum number of generation rounds. |
| | *For Today, COVID, XSum, CNN, & ExpertQA datasets:* | | |
| | Target factor count ($K$) | 40 | Target number of unique factors. |
| | Batch size ($b$) | 5 | Sentences generated per LLM call. |
| | Max rounds ($T_{\max}$) | 10 | Maximum number of generation rounds. |
| **Context-Aware Factor Mapping** | | | |
| Hierarchical Retrieval | Top-$K_1$ clusters | 3 | Number of top clusters to retrieve in the coarse search. |
| | Top-$K_2$ factors | 5 | Number of top factors to retrieve per cluster in the fine search. |
| | Prototype weight ($\alpha$) | 0.5 | Weight balancing cluster theme vs. factor content in prototypes. |
| Self-Consistent Filtering | Voting rounds ($R$) | 3 | Number of LLM calls for the majority voting mechanism. |
| | Vote ratio | 0.5 | Ratio for calculating the vote threshold $\tau$. |
| **Probabilistic Inference** | | | |
| Parameter Elicitation | Smoothing alpha ($\epsilon$) | 0.5 | The Laplace smoothing factor for CPT stabilization. |
| | LLM sampling temp. | 0.5 | Sampling temperature for LLM-based parameter elicitation. |
| | LLM parse retries | 20 | Maximum retries for eliciting valid parameters from the LLM. |
| **Fact Checking (*COVID, XSum, CNN, & ExpertQA datasets*) threshold** | | | |
| Decision-making | Probability threshold ($\tau_{\text{dec}}$) | 0.9 | Minimum probability required to label as "support." |

Table 8: Hyperparameter settings used in our experiments. Note that settings for factor generation vary by dataset group to accommodate different complexity levels, applies a decision threshold for support classification.

## A.7 SENSITIVITY TO $K$ AND METHOD COMPARISON

**Why we finally choose KNN and** ($K_1=3$, $K_2=5$)    All analyses here are conducted on the HAR-BOR model built on Qwen2.5-72b. KNN consistently delivers a low Unknown Rate and stable average F1 in our tests, avoiding BM25's high unknown proportion and FAISS's fluctuations. Very small $K$ values (2/3) under-cover relevant factors, while enlarging them to $4/8$ actually degrades performance relative to our tuned $(3, 5)$ setting because more noisy neighbors are included. Hence, the mid-range $(3, 5)$ strikes a better coverage–precision trade-off. Moreover, KNN's Euclidean distance is robust to our factor-embedding scale and requires minimal tuning and engineering overhead. **Accordingly, the main results reported in Table 1 are obtained with KNN using** ($K_1=3$, $K_2=5$)**.**

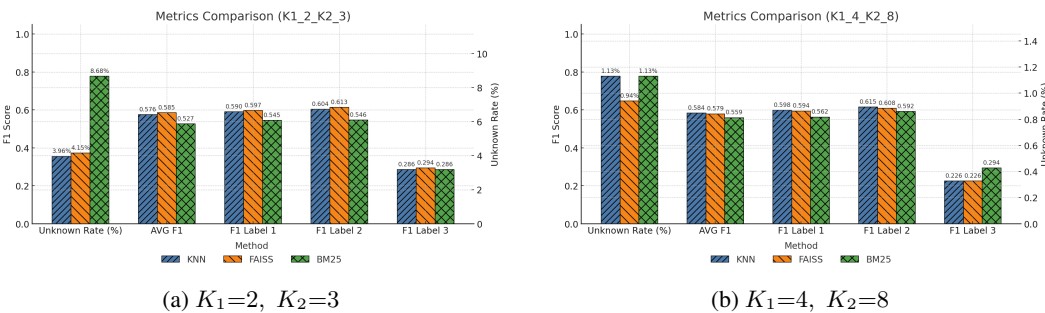

(a) $K_1=2$, $K_2=3$    (b) $K_1=4$, $K_2=8$

Figure 4: Unknown Rate and per-class F1 comparison across KNN, FAISS, and BM25 under two $K$ settings.

## A.8 FURTHER ANALYSIS: CLUSTERING AND LLM CALIBRATION BIAS

### A.8.1 CLUSTERING QUALITY AND ALGORITHM CHOICE

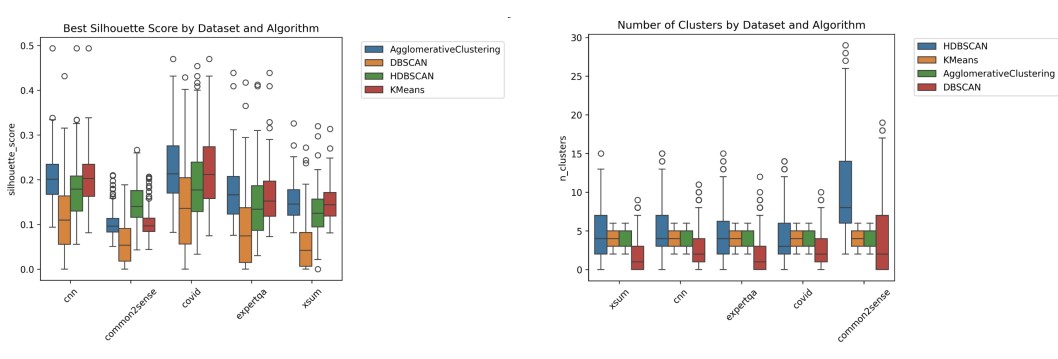

(a) Silhouette Score by Dataset and Algorithm    (b) Number of Clusters by Dataset and Algorithm

Figure 5: Comparison of clustering quality and flexibility across algorithms.

We adopt **HDBSCAN** as our clustering algorithm due to its ability to automatically determine the number of clusters based on data density, which is especially important in open-ended and semantically diverse factor spaces. As shown in Figure 5(b), HDBSCAN produces a wider and more adaptive range of cluster counts across datasets compared to other algorithms.

Although fixed-$k$ algorithms like KMeans or Agglomerative Clustering occasionally yield higher silhouette scores on certain datasets, their performance heavily depends on the manually chosen number of clusters, which may not generalize across tasks. This fixed structure risks oversimplifying or overfragmenting the semantic space when the true number of factor types varies with input complexity.

DBSCAN also determines clusters automatically, but tends to merge semantically distinct groups or produce very few clusters, especially under high noise or sparse factor conditions.

Overall, HDBSCAN provides a good trade-off between clustering quality and adaptability, making it suitable for our pipeline that operates across datasets with diverse semantic granularity.

### A.8.2 EFFECT OF PROBABILITY-ESTIMATOR LLM AND CALIBRATION

To test how sensitive HARBOR is to such calibration effects, we replace Qwen2.5-72B with the smaller GPT-4o-mini while keeping the factor space, mapping pipeline, and NB+CBN inference fixed, we evaluate on CNN, Today, Plasma, and ExpertQA; as shown in Table 9, all changes are

| Dataset | Qwen2.5-72B | GPT-4o-mini | $\Delta$ (mini $-$ 72B) |
|---|---|---|---|
| CNN | 62.9 | 62.3 | $-0.6$ |
| ExpertQA | 60.5 | 58.8 | $-1.7$ |
| Plasma | 76.2 | 73.8 | $-2.4$ |
| Today | 82.7 | 83.8 | $+1.1$ |

Table 9: Task performance (%) of HARBOR with different probability-estimator LLMs.

within about $\pm 2.5$ points, indicating limited sensitivity to the specific choice of probability-estimator LLM.

We further analyze how changing the probability-estimator LLM reshapes factor-level scores: Table 10 reports distributional distances between factor-score distributions under Qwen2.5-72B and GPT-4o-mini, and Figure 6 visualizes the corresponding smoothed factor-level probability profiles on CNN, TODAY, PLASMA, and EXPERTQA, showing that only CNN exhibits a pronounced distributional shift while overall factor patterns remain similar, consistent with the small performance differences in Table 9.

| Dataset | JS div. | Wass. dist. | KS $p$-value | Diff. |
|---|---|---|---|---|
| CNN | 0.356 | 0.074 | 0.0001 | significant |
| TODAY | 0.235 | 0.042 | 0.7012 | not significant |
| PLASMA | 0.205 | 0.040 | 0.7287 | not significant |
| EXPERTQA | 0.313 | 0.045 | 0.0773 | not significant |

Table 10: Distributional distances between factor scores under Qwen2.5-72B and GPT-4o-mini.

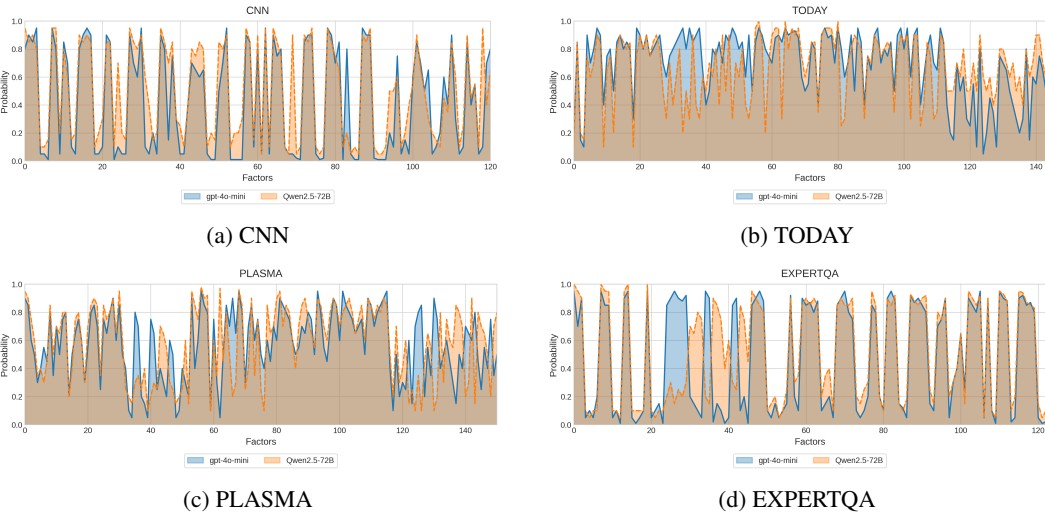

(a) CNN  (b) TODAY

(c) PLASMA  (d) EXPERTQA

Figure 6: Smoothed factor-level probability profiles under Qwen2.5-72B and GPT-4o-mini on four datasets.

### A.9  DETAILED ALGORITHM SPECIFICATIONS

This section provides the detailed pseudo-code for the three core stages of the HARBOR framework, as described in Section 4. Algorithm 1 details the construction of the hierarchical factor space. It operates by iteratively generating raw factors, validating them via self-consistency voting, and finally organizing them using a pipeline of embedding, clustering, and LLM-based thematic pruning. Algorithm 2 specifies the multi-stage process for mapping a condition to a relevant factor set. The pipeline begins with a broad, two-level hierarchical retrieval to generate candidates, followed by a voting-based filtering step and a final reflective refinement to ensure high precision. Algorithm 3

---

**Algorithm 1:** Iterative Factor Generation via Bottom-up Abduction

---

**Input:** Scenario description $s$, primary outcome $O_1$, secondary outcome $O_2$, LLM $\mathcal{M}$, target factor count $K$, batch size $b$, maximum rounds $T_{\max}$, clustering flag $C$

**Output:** Hierarchical factor space $\widetilde{\mathcal{F}}$, factor–outcome mapping $M$, clustering statistics

1   $\mathcal{F}^{(0)} \leftarrow \emptyset$;
2   **for** $t \leftarrow 1$ **to** $T_{\max}$ **do**
3      **if** $|\mathcal{F}^{(t-1)}| \geq K$ **then**
4         **break**
      // Contextual sentence generation
5      $S^{(t)} \leftarrow \text{GENERATESENTENCES}(s, O_1, O_2, b, \mathcal{M})$;
      // Factor harvesting and validation
6      $\Delta\mathcal{F}^{(t)} \leftarrow \text{EXTRACTFACTORS}(S^{(t)}, \mathcal{M})$;
7      $\mathcal{F}^{(t)} \leftarrow \mathcal{F}^{(t-1)} \cup \Delta\mathcal{F}^{(t)}$;

8   $M \leftarrow \text{VOTESUPPORT}(\mathcal{F}^{(T)}, s, O_1, O_2, \mathcal{M})$;
9   $\mathcal{F}_{\text{validated}} \leftarrow \{f \in \mathcal{F}^{(T)} : M[f] \in \{\text{``}O_1\text{''}, \text{``}O_2\text{''}, \text{``Neutral''}\}\}$;
10   **if** $C = \textit{True}$ **then**
      // Encode factors into dense representations
11      $E \leftarrow \{e_f : f \in \mathcal{F}_{\text{validated}}\}$ using MiniLM;
      // Dimensionality reduction and clustering
12      $E_{\text{reduced}} \leftarrow \text{UMAP}(E)$;
13      $\mathcal{C}_{\text{raw}} \leftarrow \text{HDBSCAN}(E_{\text{reduced}})$;
      // LLM-guided thematic organization and pruning
14      $\mathcal{C}_{\text{themed}} \leftarrow \text{ASSIGNTHEMES}(\mathcal{C}_{\text{raw}}, \mathcal{M})$;
15      $\widetilde{\mathcal{F}} \leftarrow \text{PRUNEREDUNDANCY}(\mathcal{C}_{\text{themed}}, M, \mathcal{M})$;
16      compute clustering statistics;
17   **else**
18      $\widetilde{\mathcal{F}} \leftarrow \{(\text{``default''}, \mathcal{F}_{\text{validated}})\}$;
19      set default clustering statistics;

20   **return** $(\widetilde{\mathcal{F}}, M, \textit{clustering statistics})$;

---

outlines the procedure for transforming a set of mapped factors into a final probability. It involves parameterizing two parallel models—Naïve Bayes and a latent-augmented Causal Bayesian Network—with probabilities elicited from an LLM, and then aggregating their posteriors for a robust final estimate.

## A.10   ADDITIONAL DETAILS FOR THE BIRD-ON-HARBOR ABLATIONS

**Complete-information spaces in BIRD vs. HARBOR.**   Conceptually, BIRD assumes that each factor takes two mutually exclusive attribute values. For a given scenario, these categorical factors form a low-arity product space, so the complete-information training set can be obtained by (conceptually) enumerating this Cartesian product and sampling from it. In HARBOR, factors are standalone evidence-like statements drawn from a global pool and organized into clusters and themes, not tied to a fixed two-value choice per slot. A complete-information assignment is thus a subset of this pool, and naively forming a Cartesian product over clusters would both generate many incoherent combinations and lead to a combinatorial explosion.

**Structured Monte Carlo scheme for CPT training.**   To plug BIRD's CPT-based outcome model into HARBOR's factor space, we therefore adopt a structured Monte Carlo scheme. For each mapped condition, we fix a "base" set of factors returned by the mapping pipeline, randomly complete it with a bounded number of additional factors up to a maximum length $L$, and treat the resulting complete assignments

$$\{f^{(1)}, \ldots, f^{(m)}\}$$

**Algorithm 2:** Context-Aware Factor Mapping

**Input:** Hierarchical factor space $\widetilde{\mathcal{F}} = \{(C_j, F_j)\}$, unclustered factors $F_u$, condition $u$, cluster top-$K_1$, factor top-$K_2$, voting rounds $R$, vote threshold $\tau$, balance weight $\alpha$
**Output:** High-precision factor set $\mathcal{F}^*(u)$

1   $\mathcal{F}_{\text{cand}} \leftarrow \emptyset$;
2   **if** $\widetilde{\mathcal{F}} \neq \emptyset$ *(clustered factors exist)* **then**
     // Compute cluster prototypes
3     **foreach** $(C_j, F_j) \in \widetilde{\mathcal{F}}$ **do**
4       $e_{C_j} \leftarrow \alpha \cdot \text{EMBED}(\text{theme}(C_j)) + (1 - \alpha) \cdot \frac{1}{|F_j|} \sum_{f \in F_j} \text{EMBED}(f)$;
     // Build KNN indices
5     $\text{KNN}_{\text{clusters}} \leftarrow \text{BUILDKNN}(\{e_{C_j}\}, K_1)$;
6     **foreach** $(C_j, F_j) \in \widetilde{\mathcal{F}}$ **do**
7       $\text{KNN}_{F_j} \leftarrow \text{BUILDKNN}(\{\text{EMBED}(f) : f \in F_j\}, K_2)$;
     // Coarse-to-fine retrieval
8     $C_{\text{sel}} \leftarrow \text{KNN}_{\text{clusters}}.query(\text{EMBED}(u))$;
9     **foreach** $C_j \in C_{\text{sel}}$ **do**
10      $F_{\text{retrieved}} \leftarrow \text{KNN}_{F_j}.query(\text{EMBED}(u))$;
11      $\mathcal{F}_{\text{cand}} \leftarrow \mathcal{F}_{\text{cand}} \cup F_{\text{retrieved}}$;
12   **if** $F_u \neq \emptyset$ *(unclustered factors exist)* **then**
13     $\text{KNN}_{\text{unclustered}} \leftarrow \text{BUILDKNN}(\{\text{EMBED}(f) : f \in F_u\}, K_2)$;
14     $\mathcal{F}_{\text{cand}} \leftarrow \mathcal{F}_{\text{cand}} \cup \text{KNN}_{\text{unclustered}}.query(\text{EMBED}(u))$;
15   initialize vote counts $v[f] \leftarrow 0$ for all $f \in \mathcal{F}_{\text{cand}}$;
16   **for** $r \leftarrow 1$ **to** $R$ **do**
17     $S^{(r)} \leftarrow \text{LLMSELECT}(u, \mathcal{F}_{\text{cand}}, \mathcal{M})$;
18     **foreach** $f \in S^{(r)}$ **do**
19      $v[f] \leftarrow v[f] + 1$;
20   $\mathcal{F}_{\text{vote}} \leftarrow \{f \in \mathcal{F}_{\text{cand}} : v[f] \geq \tau\}$;
21   $\mathcal{F}^*(u) \leftarrow \text{REFLECTIVEREFINE}(u, \mathcal{F}_{\text{vote}}, \mathcal{M})$;
22   **return** $\mathcal{F}^*(u)$;

as samples from a distribution $\pi$ over the intractable complete-information space. This allows us to approximate the BIRD training objective

$$\mathcal{L}(\theta) = \mathbb{E}_{f \sim \pi}\big[\ell(P_\theta(O \mid f), y(f))\big] \approx \hat{\mathcal{L}}_m(\theta) = \frac{1}{m} \sum_{i=1}^{m} \ell\big(P_\theta(O \mid f^{(i)}), y(f^{(i)})\big),$$

where $P_\theta(O \mid f)$ is the outcome model with parameters $\theta$, $\ell$ is the training loss, and $y(f^{(i)})$ is the LLM's coarse probability assessment under complete information $f^{(i)}$. In the experiments we match the original BIRD setting with $m = 128$ sampled assignments per scenario.

**Limitations and motivation for LLM-based priors.** This Monte Carlo construction is practical but imperfect: with a finite sampling budget, some factors may never appear in the training set and remain tied to their priors, and uniform completion from a large factor pool can under-cover parts of the space or induce slightly unrealistic co-occurrences. Together with prior work showing that LLMs can provide informative priors and probabilistic knowledge for Bayesian models (e.g., Gouk and Gao, 2024; Riegler et al., 2025; Nafar et al., 2025), this motivates HARBOR's main design choice of eliciting coarse factor-level probabilities directly from the LLM instead of relying purely on CPT fitting.

---

**Algorithm 3:** Probabilistic Inference with Elicited Parameters

---

**Input:** Mapped factor set $\mathcal{F}^*(u)$, outcome hypotheses $\{O_1, O_2\}$, LLM $\mathcal{M}$, aggregation weights $w_{\text{NB}}, w_{\text{CBN}}$, smoothing parameter $\epsilon$

**Output:** Aggregated posterior probability $P_{\text{agg}}(O_1 \mid \mathcal{F}^*(u))$

    `// Factor-level parameters for both models`

**1 foreach** $f \in \mathcal{F}^*(u)$ **do**

**2**    $\theta_f \leftarrow$ ELICITPOSTERIOR$(f, O_1, \mathcal{M})$ // $\theta_f \approx P(O_1 \mid f)$

**3**    $\theta_f \leftarrow$ SMOOTH$(\theta_f, \epsilon)$;

    `// Build NB model with conditional independence assumption`

**4** Build NB structure: $O \rightarrow f$ for all $f \in \mathcal{F}^*(u)$;

**5 foreach** $f \in \mathcal{F}^*(u)$ **do**

**6**    Set CPT: $P(f \mid O_1) = \theta_f$, $P(f \mid O_2) = 1 - \theta_f$;

**7** Set uniform prior: $P(O_1) = P(O_2) = 0.5$;

**8** $P_{\text{NB}} \leftarrow$ INFERPOSTERIOR$(\text{NB}, \mathcal{F}^*(u))$ using Eq. 2;

    `// Latent variable discovery and structure learning`

**9** $\mathcal{L} \leftarrow$ ELICITLATENTVARIABLES$(\mathcal{F}^*(u), \mathcal{M})$;

**10** partition $\leftarrow$ ASSIGNFACTORSTOLATENTS$(\mathcal{F}^*(u), \mathcal{L}, \mathcal{M})$;

    `// Latent-level parameter elicitation`

**11 foreach** $L_i \in \mathcal{L}$ **do**

**12**    $p_{i1} \leftarrow$ ELICITCONDITIONAL$(L_i, O_1, \mathcal{M})$ // $p_{i1} = P(L_i = 1 \mid O_1)$

       $p_{i2} \leftarrow$ ELICITCONDITIONAL$(L_i, O_2, \mathcal{M})$ // $p_{i2} = P(L_i = 1 \mid O_2)$

       $(p_{i1}, p_{i2}) \leftarrow$ SMOOTH$((p_{i1}, p_{i2}), \epsilon)$;

    `// Build CBN structure and parameterize CPTs`

**13** Build CBN structure: $L_i \rightarrow f$ for $f \in \text{partition}(L_i)$, $L_i \rightarrow O$ for all $L_i$;

**14 foreach** $L_i \in \mathcal{L}$ **do**

**15**    Set uniform prior: $P(L_i = 1) = 0.5$;

**16**    **foreach** $f \in partition(L_i)$ **do**

**17**      Set CPT: $P(f \mid L_i = 1) = \theta_f$, $P(f \mid L_i = 0) = 1 - \theta_f$;

**18** Derive outcome CPT: $P(O \mid L_1, \ldots, L_k)$ using $\{p_{i1}, p_{i2}\}_{i=1}^k$ via Bayes' rule;

**19** $P_{\text{CBN}} \leftarrow$ INFERPOSTERIOR$(\text{CBN}, \mathcal{F}^*(u))$ using variable elimination;

**20** $P_{\text{agg}} \leftarrow w_{\text{NB}} \cdot P_{\text{NB}} + w_{\text{CBN}} \cdot P_{\text{CBN}}$;

**21 return** $P_{\text{agg}}$;

---

# B APPENDIX B

## B.1 LLM USAGE STATEMENT

We clarify the role of LLMs in our research workflow. Specifically, LLMs were *only* used to assist with (i) code writing (e.g., generating boilerplate code, debugging minor errors, and improving readability), and (ii) polishing the writing of the paper text (e.g., improving clarity, grammar, and style). No experimental results, theoretical analyses, or substantive scientific claims in this paper were produced by LLMs. All methodological designs, experiments, and conclusions are solely the work of the authors.

## B.2 COMPREHENSIVE EVALUATION OF FACTOR SPACES AND MAPPINGS

In this subsection, we first compare the factor spaces generated by HARBOR and BIRD (Table 11), then examine their mapping performance under specific cooking conditions (Table 12), and finally analyze the organization of latent nodes for Outcome1 (Table 13). As shown in Table 11, HARBOR produces a more hierarchically structured set of categories that cover safety and nutrition dimensions while maintaining low redundancy, whereas BIRD's space exhibits multiple synonymous and fragmented entries. Table 12 demonstrates that HARBOR successfully maps key factors—such as food safety, temperature control, and process precision—under conditions like pathogen elimination at high temperatures, large-batch cooking, and high-altitude environments, filling gaps left by BIRD. Furthermore, Table 13 presents HARBOR's five latent dimensions for Outcome1 (TextureLat, Flavor-

Lat, EfficiencyLat, SafetyLat, and ProcessControlLat), each with a logically coherent and balanced distribution of factors, offering robust support for downstream factor analysis and visualization. Overall, the HARBOR method shows clear advantages in generating a high-cohesion, low-redundancy, and practically actionable factor space.

---

**Scenario**: The efficiency of cooking noodles is being compared between hot water and warm water.
**Outcome1**: Noodles cook much more efficiently in hot water than they do in warm water.
**Outcome2**: Noodles cook much more efficiently in warm water than they do in hot water.

| **HARBOR Generated Factor Space** | |
|---|---|
| **Clumping Prevention** | hard inside; reduced sticking; clumping reduction; rapid softening |
| **Efficiency Reductions** | energy and time efficiency; practicality; energy efficiency |
| **Texture & Flavor** | texture preservation; al dente texture; flavor; dispersing flavorings; texture inconsistency; uniform texture; longer gelatinization; palatability; better flavor infusion |
| **Cooking Efficiency** | ensures even cooking and texture retention; inefficient activation of cooking process; professional kitchens; cooking environment; slow outer layer cooking; reduced cooking time; gradual cooking; difficulty controlling cooking process |
| **Starch Digestion Efficiency** | effective breakdown of starches; starch breakdown inefficiency; digestibility; gluten breakdown |
| **Temperature Control** | inconsistent temperature control; insufficient thermal energy; lower temperature; consistent use of hot water; noodle hydration; evaporation; temperature of water; pre-soaking benefits; delicate noodles |
| **Cooking Safety & Precision** | risk of undercooking or overcooking; avoiding overcooking/mushiness; food safety (reduces risk of foodborne illness); overcooking prevention |

| **BIRD Generated Factor Space** | |
|---|---|
| **Cooking Temperature and Time** | Hot water significantly reduces cooking time by breaking down starches more quickly; Warm water requires a longer cooking time for noodles to become tender |
| **Noodle Hydration Process** | Hot water leads to faster hydration and cooking of noodles; Warm water results in a slower but more even hydration process |
| **Recommended Cooking Time** | Shorter recommended cooking time in hot water; Longer recommended cooking time in warm water |
| **Energy Efficiency** | Hot water uses more energy to heat up initially but cooks faster, potentially saving overall energy; Warm water uses less energy to heat up but requires a longer cooking time, potentially using more energy overall |
| **Texture and Quality** | Hot water can lead to a more uniform texture and prevent clumping of noodles; Warm water may result in a softer texture but can cause the noodles to become mushy if cooked for too long |
| **Cooking Efficiency** | Significantly reduces cooking time with hot water; Requires longer cooking time with warm water |
| **Noodle Texture** | Hot water ensures a more uniform texture; Warm water can lead to a softer texture |
| **Starch Breakdown** | High temperature in hot water breaks down starches more quickly; Lower temperature in warm water results in slower starch breakdown |
| **Energy Consumption** | Hot water uses more initial energy but cooks faster; Warm water uses less initial energy but cooks longer |
| **Hydration Process** | Hot water leads to faster hydration; Warm water results in a slower but more even hydration |
| **Clumping Prevention** | Hot water helps prevent noodles from clumping; Warm water may increase the risk of noodles clumping |

Table 11: An example of HARBOR and BIRD generated factor spaces

## B.3 PROMPTS

**Example Prompt for Generating Supporting/Refuting Sentences**

**System**
You are an AI assistant that helps people make decisions.
Generate {n} diverse supporting or refuting sentences for scenario:
{scenario}, comparing '{Outcome}' vs '{oppo_Outcome}'.

**User**
Scenario:  Alice is training for a marathon.
Outcome:  Running on a treadmill improves her endurance.
Opposite Outcome:  Running on a treadmill does not improve her
endurance.
Generate 2 sentences.

**Assistant**
1.  Treadmill training allows Alice to maintain a consistent pace and
monitor heart rate, boosting her aerobic capacity.
2.  The treadmill's adjustable incline simulates hill workouts,
increasing leg strength and stamina.

**User**
Scenario:  Bob studies every evening.
Outcome:  Studying in short, focused bursts enhances retention.
Opposite Outcome:  Studying in short, focused bursts does not enhance
retention.
Generate 2 sentences.

**Assistant**
1.  Brief study sessions with breaks prevent mental fatigue and improve
long-term recall.
2.  Frequent reviews in small intervals reinforce memory pathways,
aiding retention.

Figure 7: Example Prompt for Generating Supporting or Refuting Sentences

**Scenario**: The efficiency of cooking noodles is being compared between hot water and warm water.

**Outcome1**: Noodles cook much more efficiently in hot water than they do in warm water.

**Outcome2**: Noodles cook much more efficiently in warm water than they do in hot water.

| Condition | BIRD Mapped Factors | HARBOR Mapped Factors |
|---|---|---|
| **The higher temperature of hot water can accelerate the gelatinization process in starchy foods like noodles.** | Shorter recommended cooking time in hot water; High temperature in hot water breaks down starches more quickly; Hot water can lead to a more uniform texture and prevent clumping of noodles; Hot water leads to faster hydration and cooking of noodles | noodle hydration; temperature of water; effective breakdown of starches; digestibility; better flavor infusion; dispersing flavorings; flavor |
| **Cooking in hot water reduces the amount of total time that the noodles spend in the water.** | Shorter recommended cooking time in hot water; High temperature in hot water breaks down starches more quickly; Significantly reduces cooking time with hot water; Hot water leads to faster hydration and cooking of noodles; Hot water leads to faster hydration; Hot water significantly reduces cooking time by breaking down starches more quickly | reduced cooking time; ensures even cooking and texture retention; noodle hydration; temperature of water; avoiding overcooking/mushiness; overcooking prevention |
| **Hot water helps in killing any potential foodborne pathogens or microbes present in the noodles.** | None | consistent use of hot water; temperature of water; food safety (reduces risk of foodborne illness) |
| **You're cooking a large quantity of noodles.** | None | gradual cooking; inefficient activation of cooking process; ensures even cooking and texture retention; cooking environment; avoiding overcooking/mushiness; overcooking prevention; risk of undercooking or overcooking; starch breakdown inefficiency; effective breakdown of starches |
| **You are at a high altitude where water boils at lower temperatures than at sea level.** | None | temperature of water; lower temperature; insufficient thermal energy; risk of undercooking or overcooking; slow outer layer cooking; cooking environment; inefficient activation of cooking process; consistent use of hot water; evaporation; overcooking prevention; avoiding overcooking/mushiness; gradual cooking |

Table 12: An example of BIRD and HARBOR conditions factor mappings

**Scenario**: The efficiency of cooking noodles is being compared between hot water and warm water.
**Outcome1**: Noodles cook much more efficiently in hot water than they do in warm water.
**Outcome2**: Noodles cook much more efficiently in warm water than they do in hot water.

| Parent Node | Latent Node | Child Nodes (Factors) |
|---|---|---|
| Outcome1 | TextureLat | slow outer layer cooking; texture preservation; ensures even cooking and texture retention; al dente texture; reduced cooking time; overcooking prevention; uniform texture; gradual cooking; palatability; hard inside; starch breakdown inefficiency; reduced sticking; rapid softening; delicate noodles; avoiding overcooking/mushiness; gluten breakdown; longer gelatinization; texture inconsistency |
| Outcome1 | FlavorLat | noodle hydration; flavor; dispersing flavorings; better flavor infusion; digestibility; food safety (reduces risk of foodborne illness) |
| Outcome1 | EfficiencyLat | energy and time efficiency; energy efficiency; evaporation; practicality; pre-soaking benefits; consistent use of hot water; cooking environment |
| Outcome1 | SafetyLat | temperature of water; lower temperature; inconsistent temperature control; insufficient thermal energy |
| Outcome1 | ProcessControlLat | clumping reduction; effective breakdown of starches; inefficient activation of cooking process; risk of undercooking or overcooking; difficulty controlling cooking process; professional kitchens |

Table 13: An example of latent nodes organized under parent Outcome1 with their corresponding factors

**Few-Shot Examples for Factor Extraction**

**System**
Extract distinct factors from these sentences. Think step by step about what factors are mentioned, then provide your final answer as a JSON array.

**User**
Extract distinct factors from these sentences as a JSON array.
1. Treadmill training allows maintaining a consistent pace and monitoring heart rate, boosting aerobic capacity.
2. The treadmill's adjustable incline simulates hill workouts, increasing leg strength and stamina.

**Assistant**
Let me analyze these sentences to identify the key factors:
• Pace consistency (ability to maintain steady speed)
• Heart rate monitoring (tracking cardiovascular response)
• Adjustable incline (variable difficulty/terrain simulation)
• Leg strength (muscle development)
Final answer: ["Pace consistency","Heart rate monitoring", "Adjustable incline","Leg strength"]

**User**
Extract distinct factors from these sentences as a JSON array.
1. Group work requires coordination between team members.
2. Individual tasks allow for personal control and efficiency.

**Assistant**
Let me identify the key factors from these sentences:
• Coordination requirements (need for team synchronization)
• Personal control (individual autonomy)
• Efficiency (productivity/effectiveness)
• Team collaboration (working together)
Final answer: ["Coordination requirements","Personal control", "Efficiency","Team collaboration"]

Figure 8: Few-shot prompt–response pairs for factor extraction.

---

**Few-Shot Examples for Factor–Outcome Voting**

**System**
Decide which outcome the factor supports.  Reason briefly (1-2
sentences), then provide your final answer as a JSON object.  Keep
the explanation as short as possible--no extra commentary.

**User**
Scenario:  Alice trains for a marathon.
Outcome1:  Treadmill running improves endurance.
Outcome2:  Treadmill running does not improve endurance.
Factor:  Pace consistency
Decide which outcome this factor supports:  Outcome1, Outcome2, or
Both.  Think step by step, then answer in JSON.

**Assistant**
Pace consistency forces a steady speed, building aerobic capacity and
sustained effort.
Final answer:  {"Pace consistency": "Outcome1"}

**User**
Scenario:  Alice trains for a marathon.
Outcome1:  Treadmill running improves endurance.
Outcome2:  Treadmill running does not improve endurance.
Factor:  Weather conditions
Decide which outcome this factor supports:  Outcome1, Outcome2, or
Both.  Think step by step, then answer in JSON.

**Assistant**
Treadmill gives consistent conditions, yet outdoor weather readies race
adaptability.
Final answer:  {"Weather conditions": "Both"}

---

Figure 9: Few-shot prompt–response pairs for factor–outcome voting.

---

**Few-Shot Examples for Theme Name Generation**

**System**
Generate a concise English theme name (1-3 words) that captures the
common topic of these factors.
Return only the theme name, no explanation.

**User**
Generate a theme name for these related factors:
["energy expenditure", "energy transfer efficiency"]

**Assistant**
Energy Efficiency

**User**
Generate a theme name for these related factors:
["precision control", "better control (accuracy)"]

**Assistant**
Control Precision

---

Figure 10: Few-shot prompt–response pairs for generating concise theme names.

**1566**
**1567**
**1568**
**1569**
**1570**

---

**Few-Shot Examples for Factor–Condition Mapping**

**System**
You are an expert at analyzing logical relationships between conditions and factors. Given a condition and a list of candidate factors, select factors that have reasonable connections to the condition. Be inclusive rather than restrictive.

**User**
Scenario: A student is preparing for final exams.
Condition: The student spends more time in the library.
Candidate factors: ["Better time management", "More stress", "Increased social activities"]
Task: Select the factor list that is most reasonably related to the given condition. Think step by step about each factor's relevance, then provide your selection. Please respond with your reasoning followed by Final answer: and a JSON object in this exact format: {"answer": ["factor1", "factor2", ...]}.

**Assistant**
Let me analyze each factor: • "Better time management": Spending more time in the library suggests the student is organizing their study schedule better and dedicating focused time to studying. This is directly related. • "More stress": While exam preparation can be stressful, spending more time in the library is typically a positive study behavior, not necessarily indicating increased stress. • "Increased social activities": Libraries are generally quiet study spaces, not social venues. More library time would likely mean less social activities. Final answer: {"answer": ["Better time management"]}

**User**
Scenario: A company introduces remote working policies.
Condition: Employees can work from home twice a week.
Candidate factors: ["Improved work-life balance", "Reduced office costs", "More commuting"]
Task: Select the factor list that is most reasonably related to the given condition. Think step by step about each factor's relevance, then provide your selection. Please respond with your reasoning followed by Final answer: and a JSON object in this exact format: {"answer": ["factor1", "factor2", ...]}.

**Assistant**
Let me evaluate each factor: • "Improved work-life balance": Working from home twice a week allows employees to save commute time, have more flexibility, and better integrate work with personal life. This is directly related. • "Reduced office costs": With employees working from home part-time, the company needs less office space, utilities, and resources. This is a logical consequence. • "More commuting": This is contradictory--working from home twice a week would actually reduce commuting, not increase it. Final answer: {"answer": ["Improved work-life balance","Reduced office costs"]}

Figure 11: Few-shot prompt–response pairs for factor–condition mapping.

**Few-Shot Examples for Factor Reflection**

**System**
You are performing a self-reflection task. Given a condition and a list of initially selected factors, review each factor with a LENIENT approach. Keep factors that have ANY reasonable connection to the condition. Only remove factors that are clearly irrelevant or contradictory. When uncertain, err on the side of keeping the factor. Think step by step about each factor, then provide your reasoning followed by 'Final answer:' and a JSON array of the factors to keep.

**User**
Condition: City implements a bike-sharing program.
Initially selected factors: ["Increased bike usage", "Higher car sales", "More traffic jams"] Task: Review and keep factors reasonably related to the condition. Think step by step about each factor's relevance, then 'Final answer:' with a JSON array.

**Assistant**
"Increased bike usage" is the direct result of more shared bikes--keep it.
"Higher car sales" has no clear link to bike sharing--remove it.
"More traffic jams" could occur if road space shifts to bikes and cars interact--keep it.
Final answer: ["Increased bike usage", "More traffic jams"]

**User**
Condition: Students study in a quiet library.
Initially selected factors: ["Better concentration", "Distractions from phones", "Improved retention", "Reduced social interaction"] Task: Review and keep factors reasonably related to the condition. Think step by step about each factor's relevance, then 'Final answer:' with a JSON array.

**Assistant**
"Better concentration" follows from a quiet space aiding focus--keep it.
"Distractions from phones" contradicts the library's purpose--remove it.
"Improved retention" naturally arises from focused study--keep it.
"Reduced social interaction" is a likely side-effect of silence--keep it.
Final answer: ["Better concentration", "Improved retention", "Reduced social interaction"]

Figure 12: Few-shot prompt–response pairs for lenient self-reflection on factor relevance.

**Few-Shot Examples for Probability Estimation of Factor Support**

**User**
Given the scenario:  "Comparing LED bulbs vs incandescent bulbs in
home lighting."
For each of the following factor values, please estimate the
probability
(a float between 0 and 1) that it supports Outcome1 (LED bulbs are more
advantageous)
rather than Outcome2 (incandescent bulbs are more advantageous).
Return a JSON mapping.
Factor values:
["Initial cost per bulb", "Energy consumption per hour",
"Lifespan hours", "Color rendering index",
"Warm color temperature", "Instant full brightness",
"Dimmable compatibility", "Mercury content",
"Heat generation", "Availability"]

**Assistant**
Thought:  LEDs excel in low energy use, longevity, low heat output,
and no mercury, but cost more upfront, may have lower CRI, and vary in
warmth, dimmability, and availability.
Final answer:
{
"Initial cost per bulb": 0.30, "Energy consumption per hour": 0.95,
"Lifespan hours": 0.90, "Color rendering index": 0.25,
"Warm color temperature": 0.40, "Instant full brightness": 0.50,
"Dimmable compatibility": 0.35, "Mercury content": 0.85,
"Heat generation": 0.88, "Availability": 0.45
}

**User**
Given the scenario:  {scenario}
For each of the following factor values, please estimate the
probability
(a float between 0 and 1) that it supports Outcome1:  {outcome1}
rather than Outcome2:  {outcome2}.
As reference (but not absolute) here are some initial estimates:
{prior_text}
Think step by step about each factor's relation to the outcomes,
and provide your probability estimates.  Return a JSON mapping.
Factor values:
{Factor values}

Figure 13: Few-shot prompt–response pairs for estimating the probability that a factor supports one outcome over another.

**Few-Shot Examples for Latent Variable Identification with CoT**

```
System
Please perform a brief chain-of-thought (step-by-step reasoning)
 before outputting the final JSON.
You are an AI assistant tasked with identifying latent variables
and assigning each latent only factors drawn from the provided list.
Do NOT output any edges. Return a JSON object with a single field:
  latents: an array of objects, each with:
    - name: string
- factors:  array of strings (each chosen from the provided Factors
list)
Ensure the JSON parses correctly and strictly follows this schema.

User
Please identify latent variables and assign each factor to a latent.
Then return JSON with fields:
  latents: [{"name": string, "factors": [...]}, ...]
Factors:  ["Nutrition", "Vitamins", "Taste", "Convenience"]

Assistant
Thought:  Nutrition and Vitamins both relate to health aspects of food,
while Taste and Convenience relate to user enjoyment and practicality.
Final answer:
{
"latents": [
  {"name": "HealthLat",   "factors": ["Nutrition","Vitamins"]},
  {"name": "EnjoyLat",    "factors": ["Taste","Convenience"]}
]
}

User
Please identify latent variables and assign each factor to a latent.
Then return JSON with fields:
  latents: [{"name": string, "factors": [...]}, ...]
Factors:  ["Usability", "Security", "Maintainability", "Portability",
"Reliability"]

Assistant
Thought:  Usability and Portability focus on user experience and
access, Reliability and Maintainability focus on software quality over
time, and Security is a distinct concern.
Final answer:
{
"latents": [
  {"name": "UXLat",       "factors": ["Usability","Portability"]},
  {"name": "QualityLat",  "factors": ["Reliability","Maintainability"]},
  {"name": "SecurityLat", "factors": ["Security"]}
]
}
```

Figure 14: Few-shot prompt–response pairs for latent variable identification with chain-of-thought reasoning.

**Few-Shot Examples for Latent Probability Estimation**

**System**
```
Please perform a brief chain-of-thought (step-by-step) before
outputting the final JSON:
You are an AI assistant. You will be given:
  • A list of latents where each latent has:
    - its name
    - the list of factor descriptions it groups
  • Two competing outcomes (Outcome1 vs. Outcome2)
Your task:  Think through the semantic content of the factors relative
to the outcomes,
and estimate for each latent a probability pair [p1, p0]:
  • p1 = probability the latent supports Outcome1
  • p0 = probability the latent supports Outcome2
Begin your answer with "Thought:" to show your reasoning, then output
exactly a JSON object
mapping each latent name to its [p1, p0] (no extra text).
```

**User**
```
Latents with factors:
[
"name":"Performance", "factors":["Faster processing","Efficient
resource use"],
  {"name":"Stability",   "factors":["Crash reports","Memory leaks"]}
]
Outcome1: The system improves performance.
Outcome2: The system does not improve performance.
```

**Assistant**
```
Thought:  The Performance latent groups factors that directly indicate
faster and more efficient operation, which strongly supports Outcome1.
The Stability latent lists issues that undermine reliability, which
indirectly suggests performance might not improve overall.  I'd
estimate Performance around [0.85, 0.15] and Stability around [0.30,
0.70].
Final answer:
{
  "Performance": [0.85, 0.15],
  "Stability":   [0.30, 0.70]
}
```

**User**
```
Latents with factors:
[
  {"name":"HealthLat","factors":["Nutrition benefits"]},
  {"name":"EnjoyLat", "factors":["Taste appeal","Fun presentation"]}
]
Outcome1: Healthy eating is fun.
Outcome2: Healthy eating is not fun.
```

**Assistant**
```
Thought:  Nutrition benefits relate to health but don't guarantee fun,
so HealthLat feels neutral to slightly positive.  EnjoyLat clearly
centers on taste and fun aspects, strongly supporting Outcome1.  I'd
give HealthLat [0.55, 0.45] and EnjoyLat [0.85, 0.15].
Final answer:
{
  "HealthLat": [0.55, 0.45],
  "EnjoyLat":  [0.85, 0.15]
}
```

Figure 15: Few-shot prompt–response pairs for latent probability estimation.

