# OpenReview forum: "HARBOR: Hierarchical Abduction with Bayesian Orchestration for Reliable Probability Inference in Large Language Models"
_ICLR.cc/2026/Conference — ICLR 2026 Conference Desk Rejected Submission_

### Official Review · Reviewer_NUoW · 2025-10-23

**Soundness:** 2
**Presentation:** 3
**Contribution:** 2
**Rating:** 4
**Confidence:** 3

**Summary:**

This paper proposes HARBOR, a framework for rigorous Bayesian-like inference using LLMs in decision-making scenarios. Following BIRD (Feng et al. 2025), HARBOR similarly aims to generate factors that Bayesian decision-making needs, maps a downstream condition to these factors, and estimates conditional probability tables to compute a Bayesian-constrained probability for a decision to take under the downstream condition. Compared with BIRD, HARBOR proposes three main improvements. 1) HARBOR proposes an iterative and hierarchical factor generation step to acquire more comprehensive and non-overlapping factors. 2) HABOR proposes a factor mapping (from conditions) step using hierarchical retrieval, instead of the entailment setup in BIRD. Both changes aim to address the drawback in BIRD where many predictions are "unknown" due to failures in mapping factors. 3) HARBOR directly elicits conditional probabilities from LLMs instead of learning them based on strong entailments and uses an additional causal Bayesian network (CBN) to facilitate the decision.

Experiments show that HARBOR largely resolves the "unknown" issue presented in BIRD, and outperforms BIRD in decision-making evaluation settings, while reducing the generation costs.

**Strengths:**

The problem setup, following BIRD, is sound, and HARBOR addresses a main issue in BIRD, which is the failure in factor mapping. The proposed hierarchical factor generation could inspire future research and industrial applications. The empirical results are strong and support the main claims.

**Weaknesses:**

While this paper is nicely executed, it essentially follows the BIRD setup for the general approach and evaluation. The main differences (as stated in the summary) do make a significant empirical difference, especially in coverage; however, they seem to be mostly engineering efforts that have limited contribution to future research. It seems to me that it was not BIRD's point to make every engineering effort, but to demonstrate the applicability and learnability of Bayesian-like inferences.

At the same time, I am not sure if some of the design choices are sound. For example, directly eliciting conditional probabilities from the LLMs defeats the purpose of having an external Bayesian inference in the first place, since we would be largely trusting the model's estimations of probabilities, and it is hard to tell if the gains are from more capable/calibrated models or from better inference approaches. At the same time, mapping factors from conditions using embedding-based retrieval may further decrease the trustworthiness of model inferences, especially when downstream performances are not the top priority in high-stakes setups (in other words, saying "unknown" or abstention may sometimes be preferred).

The authors could also better describe key differences with BIRD. The term "reverse abduction" is a little misleading since BIRD already employed somewhat of a "reverse" abduction in first generating scenarios, and then the factors.

**Questions:**

1. It seems that the reported BIRD performances in Table 1 are lower than those reported in the original BIRD paper, especially since BIRD is re-implemented with a larger, more capable LLM. Could you please elaborate a little on what that is? I understand that the larger dataset and forcing factors could be a difference, but for the 72B version (99% coverage), should it be closer?

---

> ### Author Response · Authors · 2025-11-16
>
> We sincerely thank reviewer for taking the time to carefully read our paper and for the thoughtful, constructive feedback. We are grateful that you find the problem setup sound, recognize that HARBOR effectively addresses BIRD’s factor-mapping issue, and see potential in our hierarchical factor generation and empirical results. At the same time, we take your concerns about conceptual novelty beyond BIRD, the implications of directly eliciting probabilities and using embedding-based retrieval (especially when abstention may be preferable), the “reverse abduction” terminology, and the reported BIRD numbers in Table 1 very seriously. Below, we respond to these points in detail and explain the clarifications.
> > ### W1: While this paper is nicely executed, it essentially follows the BIRD setup for the general approach and evaluation.
>
> Thank you for this comment. We intentionally keep BIRD’s overall setup so that improvements can be attributed to design rather than changed tasks, but within this shared framework HARBOR introduces three ingredients that we see as useful building blocks for larger-scale decision systems: an iteratively constructed **hierarchical factor space** that is designed to organize much larger factor pools, a **structured retrieval–based mapping** procedure with self-consistency and reflection instead of local entailment checks, and an **external Bayesian layer** that combines a Naïve Bayes backbone with a latent CBN while using LLM probabilities only as coarse, interpretable local signals.
>
> Our additional experiment reported to reviewer Kue9, where we swap Qwen2.5-72B for the smaller GPT-4o-mini but keep the factor space, mapping, and NB+CBN inference fixed, show that performance is mainly sensitive to this structure and only mildly to the choice of LLM for probability elicitation, indicating that the gains are not simply due to a stronger or better-calibrated backbone. On the evaluation side, current datasets only provide instance-level labels, so a natural next step that HARBOR explicitly motivates is to construct richer benchmarks with factor-level and probability or utility annotations, where hierarchical factor spaces and external Bayesian layers can be more fully assessed and further developed.
>
>
>
>
>
> > ### W2: however, they seem to be mostly engineering efforts that have limited contribution to future research. It seems to me that it was not BIRD's point to make every engineering effort, but to demonstrate the applicability and learnability of Bayesian-like inferences.
>
> Thank you for raising this concern. We agree that BIRD’s main goal was to demonstrate the basic applicability and learnability of Bayesian-like inference, and we deliberately keep its overall setup to build directly on that line of work. Our aim with HARBOR, however, is not merely to refine BIRD’s implementation, but to surface and validate three design patterns that we believe are broadly useful for future LLM–Bayes systems. First, we adopt a **factor-space-first, hierarchical design**: factors are constructed via an iterative “sentences → factors → clusters/themes” process and organized into a two-level hierarchy with explicit coverage–precision trade-offs. This turns the factor space into a scalable knowledge layer that can support much larger outcome spaces and factor pools than the flat, mutually exclusive factors in BIRD, which is crucial for more complex or domain-specific decision systems. Second, we **reformulate mapping as structured retrieval** rather than local entailment: prototype-based cluster retrieval, self-consistency voting, and reflection-based pruning together provide a reusable recipe for integrating retrieval-style evidence selection with probabilistic reasoning, and can naturally inform future RAG–Bayes hybrids where retrieval quality and robustness are central bottlenecks. Third, we propose a **dual-model external Bayesian layer** that couples a conservative Naïve Bayes backbone with a latent CBN over the same factor space, using the LLM only for coarse local probabilities; this offers a template for combining simple aggregators with learned causal structure in settings where one may want to swap in different causal graphs or priors. Our ablations (NB-only, CBN-only, w/o hierarchy, w/o clustering, w/o pe-LLM) and the LLM-swap experiments show that each of these structural choices materially affects performance, and that the gains are not explained by a stronger or better-calibrated backbone alone. We thus view HARBOR less as a collection of engineering tweaks to BIRD and more as a step toward a base design for future, larger-scale and higher-stakes Bayesian decision systems built on top of LLMs.

---

> ### Author Response · Authors · 2025-11-16
>
> > ### W3: At the same time, I am not sure if some of the design choices are sound. For example, directly eliciting conditional probabilities from the LLMs defeats the purpose of having an external Bayesian inference in the first place, since we would be largely trusting the model's estimations of probabilities, and it is hard to tell if the gains are from more capable/calibrated models or from better inference approaches.
>
> Thank you for raising this concern, we were mindful of exactly this issue when designing HARBOR. In our setup, the LLM is not expected to produce finely calibrated probabilities that would substitute for the Bayesian layer, but only to perform basic semantic discrimination and provide coarse, interpretable signals such as “how likely this factor supports Outcome 1 vs. Outcome 2” or a simple latent pair. NB and the CBN then treat these as rough strength indicators and combine them under an explicit external probabilistic structure. Empirically, as detailed in our response to reviewer Kue9, when we replace Qwen2.5-72B with the smaller GPT-4o-mini only for factor-level and latent probability elicitation, end-to-end performance on CNN, TODAY, PLASMA, and EXPERTQA changes only slightly. Together with our “w/o pe-llm” ablation, where coarse three-level heuristic priors already outperform direct LLM baselines, this suggests that HARBOR’s gains do not rely on fine-grained probability calibration of a particular backbone, but on the external Bayesian layer’s ability to robustly exploit coarse patterns of semantic support at the factor and latent levels.
>
>
>
> > ### W4: At the same time, mapping factors from conditions using embedding-based retrieval may further decrease the trustworthiness of model inferences, especially when downstream performances are not the top priority in high-stakes setups (in other words, saying "unknown" or abstention may sometimes be preferred).
>
> We agree that in high-stakes settings trustworthy abstention can be preferable to maximizing coverage, and HARBOR is explicitly designed to support such conservative behavior. In our framework, the system can output “unknown” whenever the mapped factor set is empty or when the maximum posterior over outcomes falls below a user-chosen decision threshold $\tau$, so whether a final decision is made is controlled directly by this threshold. Moreover, mapping is not a single embedding lookup but a constrained, audited pipeline: KNN retrieval is limited to cluster prototypes in the two-level hierarchy, we require self-consistency over multiple passes before accepting a factor, and a reflection step removes clearly irrelevant or contradictory factors. In higher-risk domains, one can retain this architecture and simply tighten these controls—for example by making the mapping stricter and raising $\tau$ —so that abstention becomes more frequent whenever evidence is weak or ambiguous.
>
>
>
> > ### W5: The authors could also better describe key differences with BIRD. The term "reverse abduction" is a little misleading since BIRD already employed somewhat of a "reverse" abduction in first generating scenarios, and then the factors.
>
> We agree that the differences to BIRD and the meaning of “reverse abduction” should be explained more clearly. In BIRD, given a scenario, the LLM directly proposes a flat list of factors, and each factor is tied to exactly one pair of mutually exclusive attribute values for the two outcomes, with no further structuring beyond this binary factor set. In HARBOR, “reverse abduction” refers instead to reversing the *abstraction direction* inside factor construction: we first iteratively generate many concrete, evidence-like sentences around the scenario and outcomes, then extract candidate factors from these sentences, and only then cluster and theme them into a hierarchical factor space that is not restricted to a single mutually exclusive attribute pair and can support richer outcome aspects and dependencies. We will refine the corresponding section to make this bottom-up, hierarchy-building process and its contrast to BIRD’s one-shot, binary, flat factor generation explicit, so that the key differences with BIRD are clear.

---

> ### Author Response · Authors · 2025-11-16
>
> >### Q1:It seems that the reported BIRD performances in Table 1 are lower than those reported in the original BIRD paper, especially since BIRD is re-implemented with a larger, more capable LLM. Could you please elaborate a little on what that is? I understand that the larger dataset and forcing factors could be a difference, but for the 72B version (99% coverage), should it be closer?
>
> As you already pointed out, both the **larger dataset** and the way we **force factors** are indeed the main reasons why our BIRD numbers are lower than those in the original paper, even with a stronger LLM.
>
> On the dataset side, we evaluate on a larger and slightly harder version of the benchmark and we keep **all** instances in the evaluation, including cases where BIRD’s factor mapping fails. In the original BIRD paper, any example whose condition cannot be mapped to at least one factor is marked as “unknown” and removed from the reported metrics; accuracy and F1 are computed only on those examples where BIRD successfully maps to some factor values.
>
> On the factor side, our “72B, 99% coverage” variant reaches high coverage by a **minimal** forced matching: we only require that each condition be mapped to at least one factor, and in most previously 0-match cases this means adding exactly one weakly related factor in a very sparse factor space. This improves coverage but can lower BIRD’s accuracy, because these extra factors are often noisy and BIRD has no way to repair the factor set.
>
> So the lower BIRD scores in Table 1 mainly reflect that we evaluate it on a larger, more challenging dataset **without discarding unknown cases**, under a conservative forced-matching scheme. Within this stricter and more uniform setting, the improvements of HARBOR over BIRD come from differences in factor-space design, mapping, and inference rather than from backbone strength or special treatment of the data.

---

> ### Comment · Reviewer_NUoW · 2025-11-21
> **Thank you for your rebuttal**
>
> I appreciate the authors for taking the time to write the rebuttal. From my perspective, all my original evaluations were correct, and the rebuttal does not add new information to my understanding. My original judgments were based on correctly recognizing the contributions (as the authors have reiterated). There are two additional discussions I would like to follow up on before I decide on my final rating.
>
> 1. The authors mentioned that a key difference between HARBOR and BIRD is that BIRD "produces a flat list of factors" whereas HARBOR "first iteratively generates many concrete, evidence-like sentences". I believe BIRD **does** generate concrete scenarios before producing the factors, making this not so much of a difference; hence, the HARBOR paper is indeed "misleading" when claiming contributions of "reverse abductions." I hope the authors can clarify precisely what is different here and ensure that all contributions are well supported.
> 2. Regarding the comparison with BIRD. While I agree that forcing BIRD outputs to all factors can showcase HARBOR's effectiveness through better coverage (to my understanding, this means a better and more comprehensive set of factors), such an experiment is insufficient for readers to understand how the other contributions (e.g., factor mapping and the way to compute probabilities) play a part. BIRD should be compared in several different ablations (e.g., with HARBOR's generated factors; with HARBOR's generated factors AND mapping) to people to really understand how much contribution HARBOR is really making. Otherwise, it seems to me that *only* the contribution of a better factor generation is substantiated, and for all other components, we could have just followed BIRD.

---

> ### Author Response · Authors · 2025-11-23
>
> We thank the reviewer for carefully revisiting our rebuttal and for clearly articulating the remaining concerns before making a final decision. We appreciate the opportunity to further clarify our contributions and their relation to BIRD, and we address the two follow-up questions (Q1–Q2) in detail below.
>
> > ### Q1: I hope the authors can clarify precisely what is different here and ensure that all contributions are well supported.
>
> We thank the reviewer for raising this point and for encouraging a clearer explanation of how HARBOR relates to BIRD.
>
> To help readers better understand the relation between HARBOR and BIRD, we will refine our wording around “reverse abduction”. In **BIRD**, the reasoning pattern can be summarized as *scenario → sentence → factor → attribute*: after generating sentences, BIRD abstracts them into factors and then assigns each factor two mutually exclusive attribute values, yielding a structured set of two-valued factors.
>
> In **HARBOR**, what we intended by “reverse” is not a different direction at the sentence level, but a **bottom-up way of structuring the factor space**. Concretely, our pipeline can be viewed as *scenario → sentence → attribute → factor (cluster) → latent*: we first collect many attribute-level cues from concrete, evidence-like sentences, then cluster them into factors, and finally connect these factors to latent nodes in the CBN. Rather than starting from a fixed list of factors and assigning attributes, we start from fine-grained attribute evidence and build up a hierarchical factor/latent structure.
>
> To avoid potential confusion, we will replace the term “reverse abduction” with a more precise description such as **“bottom-up abduction”** and explicitly clarify this distinction from BIRD in the revision. The usefulness of this design is supported by our ablations (w/o cluster, w/o hierarchy), and further corroborated by the additional ablation results we present in response to Q2, ensuring that each of the claimed contributions is empirically grounded.
>
>
>
> > ### Q2: BIRD should be compared in several different ablations (e.g., with HARBOR's generated factors; with HARBOR's generated factors AND mapping) to people to really understand how much contribution HARBOR is really making.
>
>
>
> We appreciate the reviewer’s request for a more fine-grained comparison that separates the impact of factor generation, mapping, and probability computation. In this section, we present the requested ablations that plug BIRD into HARBOR’s factor space and mapping, and we discuss how these results quantify the additional contributions of HARBOR beyond a better factor generator.
>
>
> ### 1. Ablation setup: plugging BIRD into HARBOR’s factor space
> For the ablations where we plug BIRD into HARBOR’s factor space and mapping, and to keep the comparison faithful, we do not use HARBOR’s direct factor-level probability elicitation. Instead, we run the original BIRD-style CPT training procedure on top of HARBOR’s  factor space and mappings, using the same number of sampled complete-information assignments per scenario as in the BIRD.
> ### 2. Conceptual difference in complete-information spaces
> Conceptually, BIRD assumes that each factor takes  two mutually exclusive attribute values. For a given scenario, these categorical factors form a low-arity product space, so the complete-information training set can be obtained by enumerating this Cartesian product and sampling from it. In HARBOR, by contrast, factors are standalone evidence-like statements drawn from a global pool and organized into clusters and themes, not tied to a fixed two-value choice. A complete-information assignment is therefore a subset of this pool, and a naive Cartesian product over clusters would both generate incoherent combinations and cause a combinatorial explosion.
> ### 3. Structured Monte Carlo scheme for BIRD-style CPT training
>
> We instead use a structured Monte Carlo scheme: for each mapped condition we fix a “base” set of factors from the mapping pipeline, randomly complete it with a bounded number of extra factors up to length $L$, and treat the resulting complete assignments $\{f^{(1)},\dots,f^{(m)}\}$ as samples from a distribution $\pi$ (with mass function $\pi(f)$) over the intractable complete-information space. This lets us approximate the BIRD training objective $\mathcal{L}(\theta) = \mathbb{E}\_{f \sim \pi}[\ell(P\_\theta(O \mid f), y(f))] \approx \hat{\mathcal{L}}\_m(\theta) = \frac{1}{m} \sum\_{i=1}^m \ell(P\_\theta(O \mid f^{(i)}), y(f^{(i)}))$, where $P\_\theta(O \mid f)$ is the outcome model with parameters $\theta$, $\ell$ is the training loss, and $y(f^{(i)})$ is the LLM’s coarse probability assessment under complete information $f^{(i)}$.

---

> ### Author Response · Authors · 2025-11-23
>
> ### 4. Limitations of Monte Carlo CPT training and motivation for LLM-based priors
> This Monte Carlo construction is practical but imperfect: with a finite sampling budget, some factors may never appear in the training set and remain tied to their priors, and uniform completion from a large factor pool can under-cover parts of the space or induce slightly unrealistic co-occurrences. Together with prior work by Gouk & Gao (2024), Riegler et al. (2025), and Nafar et al. (2025), which shows that LLMs can provide informative priors and probabilistic knowledge for Bayesian models, this motivates HARBOR’s main design choice of eliciting coarse factor-level probabilities directly from the LLM. Our experiments show that this LLM-based parameterization greatly reduces complete-information sampling and CPT-related LLM calls, while preserving or even improving downstream decision performance.
>
> ### 5. BIRD-on-HARBOR ablation configurations and cost reporting
> We run the original BIRD pipeline on HARBOR’s factor space and compare four variants of the mapping and aggregation components (result1). All configurations share the same HARBOR-generated factors and BIRD-style CPT training (hence “Needs CPT training = Yes” in every row). To isolate the impact of the mapping strategy and Bayesian layer, the reported cost metrics (“Avg tokens”, “Avg time”, and their normalized ratios) are measured only for the mapping stage, with factor generation and CPT fitting treated as shared.
>
> **Experimental settings.** The four configurations are:
>
> - **BIRD w/ force mapping (baseline)**: Original BIRD pipeline with force-based factor–condition mapping and Naïve Bayes, trained via CPT learning.
> - **BIRD w/ HARBOR’s generated factors**: Same as the baseline, but replacing BIRD’s factor space with HARBOR’s generated factors while keeping BIRD mapping and Naïve Bayes.
> - **BIRD w/ HARBOR’s factors + mapping**: Uses HARBOR’s factor space and HARBOR’s retrieval-based mapping to select factors for each condition, while retaining BIRD’s Naïve Bayes and CPT training.
> - **BIRD w/ HARBOR’s factors + mapping + CBN agg.**: Builds on the previous setting (HARBOR factors + HARBOR mapping) and augments BIRD’s Naïve Bayes with an additional CBN layer, combined via a linear opinion pool under the same CPT training.
>
> Result 1 shows that HARBOR’s gains are not just from a richer factor set. Running vanilla BIRD on HARBOR’s factors improves performance but greatly increases token usage and runtime, whereas adding HARBOR’s mapping recovers efficiency while still outperforming the BIRD baseline. With CBN aggregation on top of HARBOR’s factors and mapping, we obtain the strongest variant, improving accuracy and coverage at essentially the same reduced cost. Together with experiments where CPT training is replaced by LLM-based factor-level probability elicitation, this indicates that factor-space construction, mapping, CBN aggregation, and LLM-based parameterization are all substantive contributions to both performance and efficiency.
>
>
>
> Result 1: BIRD on HARBOR’s factors and mapping
> | Setting                                       | Context1 | Context2 |  Same |      Average      | Coverage | Avg tokens | Avg time (s) | Tokens/baseline | Time/baseline | Needs CPT training |
> |-----------------------------------------------|---------:|---------:|------:|------------------:|---------:|-----------:|-------------:|----------------:|--------------:|:------------------:|
> | BIRD w/ force mapping (baseline)              |   0.567  |   0.520  | 0.307 |            0.513  |  99.99%  |   454,747  |      4,258   |           1.000 |         1.000 |        Yes         |
> | BIRD w/ HARBOR’s generated factors            |   0.542  |   0.558  | 0.186 |  0.532 ↑0.019     |  99.12%  | 1,355,740  |     12,220   |  2.981 ↑1.98    |  2.870 ↑1.87  |        Yes         |
> | BIRD w/ HARBOR’s factors + mapping            |   0.532  |   0.551  | 0.156 |  0.526 ↑0.013     |  98.81%  |    75,917  |      2,344   |  0.167 ↓0.83    |  0.550 ↓0.45  |        Yes         |
> | BIRD w/ HARBOR’s factors + mapping + CBN agg. |   0.571  |   0.607  | 0.222 |  0.568 ↑0.055     |  98.81%  |    75,917  |      2,344   |  0.167 ↓0.83    |  0.550 ↓0.45  |        Yes         |
>
>
> **References**
>
> - Gouk, Henry, and Boyan Gao. "Automated prior elicitation from large language models for Bayesian logistic regression." *The 3rd International Conference on Automated Machine Learning*. 2024.
>
> - Riegler, Michael A., et al. "Using Large Language Models to Suggest Informative Prior Distributions in Bayesian Statistics." *arXiv preprint arXiv:2506.21964* (2025).
>
> - Nafar, Aliakbar, et al. "Extracting Probabilistic Knowledge from Large Language Models for Bayesian Network Parameterization." arXiv preprint arXiv:2505.15918 (2025).

---

> > ### Comment · Reviewer_NUoW · 2025-11-25
> >
> > I thank the authors for providing a detailed response. I think it is safe to conclude that BIRD does almost the same thing conceptually when generating factors, but I recognize slight differences in actual implementation. The authors' provided experiments are helpful to understand what differences HARBOR is making, and it does seem that some components, such as the CBN, work well; however, other improvements from HARBOR-generated factors and mapping are limited.
> >
> > Overall, I am willing to increase my rating to reflect that I am fine with the paper being accepted. However, I still maintain that this paper makes an incremental contribution.

---

### Official Review · Reviewer_mzSy · 2025-10-29

**Soundness:** 3
**Presentation:** 2
**Contribution:** 3
**Rating:** 6
**Confidence:** 3

**Summary:**

The authors introduce a framework for Bayesian inference making. They employ an LLM to abductively elicit latent factors, synthesize a causal Bayesian network of these factors, and use the estimated likelihoods of these factors to deliver calibrated probability estimates for competing hypotheses. Their method delivers stronger performance at lower LLM inference costs than baselines on a range of probabilistic reasoning tasks.

**Strengths:**

The introduction of a causal graph between factors allows this approach to consider a wider space of factors without systematically degrading performance.

The approach delivers stronger performance than other probabilistic reasoning systems on a variety of datasets, while delivering higher coverage and lower token costs.

**Weaknesses:**

The system appears to be intricately engineered, which could make it brittle to minor variations in the problem constructions or the LLM engines which power step-wise inferences. It is not clear whether it is possible to benchmark or optimize a LLM’s performance on a given sub-component of the pipeline.

Some elements of the presentation were unclear or unpolished:

Line 336-7: “We conduct all experiments with three models”, but four models are listed. It might be better to specify that these are three different *language* models, since “model’ is used elsewhere to describe probabilistic models

Certain terms are defined for unclear reasons and do not make the formal description more clear, e.g. Φ(u)

My primary concerns are with the paper’s presentation / clarity, and I believe it would deserve a better score if these concerns / questions were addressed.

**Questions:**

Why is the term Φ(u) = F*(u) introduced, when it is not referenced until the appendix? It does not make these expressions any more concise.


Why include the Naive Bayes model at all, if the causal Bayes Net is a more faithful formal representation of the questions that are being modeled? Is there an experiment that suggests this simpler model’s inclusion improves performance?

What advantages of HARBOR make it more token efficient than BIRD? What datasets / conditions are the cost comparisons in Figure 3 made under?

---

> ### Author Response · Authors · 2025-11-16
>
> We are grateful to the reviewer for the careful reading of our paper and the thoughtful feedback. We appreciate your positive assessment of our use of a causal factor graph to safely expand the factor space, as well as your recognition that HARBOR attains stronger performance with higher coverage and lower token costs than prior probabilistic reasoning systems across multiple datasets. At the same time, we value your concerns about the engineered nature of the pipeline, its potential brittleness to changes in task formulation or underlying LLMs, and several clarity issues in the presentation. Below, we address each of these questions and concerns in turn and indicate the revisions and clarifications we will make in the revised version.
>
> > ### W1: The system appears to be intricately engineered, which could make it brittle to minor variations in the problem constructions or the LLM engines which power step-wise inferences.
>
> - We appreciate the concern about potential brittleness. HARBOR uses a simple three-stage pipeline—(i) factor generation, (ii) context–factor mapping, and (iii) probabilistic inference—where each block is explicitly designed for fault tolerance: clustered factor prototypes make retrieval robust to noisy factors, hierarchical retrieval with voting over paraphrased prompts plus a brief reflection step reduces sensitivity to prompt wording, and a coarse CBN over a few high-level nodes is linearly combined with a Naïve Bayes backbone that provides a conservative fallback when the learned structure is imperfect. As shown in Table 3, removing individual components leads to gradual rather than catastrophic degradation, indicating that the system is structured yet not brittle to minor variations in problem construction or LLM behavior.
>
> > ### W2: It is not clear whether it is possible to benchmark or optimize a LLM’s performance on a given sub-component of the pipeline.
>
> - Each block can in principle be evaluated independently: factor generation and factor–outcome labeling against annotated ground truth (e.g., precision/recall/accuracy), context–factor mapping as a retrieval task (e.g., precision@K, recall), and probabilistic inference using calibration and discrimination metrics (e.g., Brier score, AUC) on held-out scenarios. In this paper we focus on end-to-end performance and ablations due to space constraints, and view a more systematic component-wise optimization study as important future work.
>
> > ### W3: Line 336-7: “We conduct all experiments with three models”, but four models are listed...
>
> - Thank you for pointing this out, and we apologize for the confusion. In the final version, we will correct the sentence to: “We conduct all experiments with four LLMs: Qwen2.5-32b, Qwen2.5-72b , DeepSeekV3-671b , and GPT-4 .”
>
> > ### W4: Certain terms are defined for unclear reasons and do not make the formal description more clear, e.g. Φ(u)...
>
> - Thank you for pointing this out. We agree that introducing $\Phi(u)$ on top of $F^\star(u)$ does not add clarity. In the final version, we will remove $\Phi(u)$ from the main text and state abstention directly in terms of $F^\star(u)$. We will define $\Phi$ locally there and explicitly tie it to the concrete mapping pipeline in the appendix.

---

> ### Author Response · Authors · 2025-11-16
>
> > ### Q1:  Why is the term Φ(u) = F*(u) introduced, when it is not referenced until the appendix? It does not make these expressions any more concise.
>
> Same as W4
>
> > ### Q2:Why include the Naive Bayes model at all, if the causal Bayes Net is a more faithful formal representation of the questions that are being modeled? Is there an experiment that suggests this simpler model’s inclusion improves performance?
>
>  We include the Naïve Bayes (NB) component not as a redundant add-on to the CBN, but as a simple backbone that aggregates factor evidence in a very direct and predictable way when the learned causal structure may be imperfect. The CBN is more expressive when its structure and tables are well captured, but in our setting these are induced from LLM outputs, so we explicitly combine two complementary views: NB’s straightforward factor aggregation and the CBN’s structured dependencies. In our ablations, the existing **“w/o aggregation”** variant is exactly an **NB-only** model, and we have additionally included a **CBN-only** variant. Across both backbones, each of these single-model variants performs worse than the full NB+CBN aggregation, which consistently achieves the best F1, indicating that NB contributes useful complementary information rather than being a purely cosmetic addition. The detailed results are reported in our W1 response to reviewer Kue9.
>
> > ### Q3: What advantages of HARBOR make it more token efficient than BIRD? What datasets / conditions are the cost comparisons in Figure 3 made under?
>
> The main reason HARBOR is more token efficient is the way it handles condition–factor mapping. BIRD asks the LLM to check entailment between every condition sentence and every factor value, often multiple times per pair and with an additional reflection step, so a large factor space leads to many LLM calls and long prompts. HARBOR instead organizes factors into a hierarchical space and first uses this structure to retrieve a small candidate subset for each condition, so only this reduced set is sent to the LLM. This hierarchical pruning greatly reduces both the number of calls and the tokens used per instance.The results in Figure 3, obtained on the Common2Sense dataset using the Qwen2.5-72B LLM, are based on the same experimental setup as the main experiments in Table 1, and this will be noted in the revision.

---

> > ### Comment · Reviewer_mzSy · 2025-11-19
> >
> > Thank you for the followups and clarifications. I will keep my score.

---

> ### Author Response · Authors · 2025-11-19
>
> We thank the reviewer again for the careful reading and for highlighting the importance of the paper’s presentation and clarity. As you wrote, *“My primary concerns are with the paper’s presentation / clarity, and I believe it would deserve a better score if these concerns / questions were addressed.”* We take this concern very seriously. In the revised manuscript, we have (i) corrected the description of the LLMs used (now explicitly stating that we use four LLMs), (ii) removed the redundant notation $\Phi(u)$ from the main text to avoid confusion and instead express everything directly in terms of $\mathcal{F}^*(u)$, and (iii) clarify in the ablation section that the w/o cbn and w/o nb variants correspond to using only the Naïve Bayes or only the Causal Bayesian Network at aggregation time, respectively. These changes are highlighted in the updated PDF to make them easy to inspect. We hope these edits address your concerns about clarity; if there are any remaining parts of the exposition that you still find confusing, we would be very grateful if you could point them out. We will also continue to refine the presentation in light of your and the other reviewers’ suggestions to further improve the overall quality of the paper.

---

### Official Review · Reviewer_uGGc · 2025-10-30

**Soundness:** 3
**Presentation:** 2
**Contribution:** 3
**Rating:** 6
**Confidence:** 3

**Summary:**

This paper proposes `HARBOR`, a framework for generating reliable probability estimates from LLMs. It aims to solve two problems with _prior abductive_-Bayesian methods: 1) the creation of sparse factor spaces, which leads to "unknown" predictions, where the model abstains from making a prediction, and 2) the violation of the naïve Bayes independence assumption.

1. The contribution is based on _"reverse abduction"_, which iteratively generates factors from sentences to build a dense and hierarchical factory space, which is then clustered. The idea is to reduce "unknown" (unmapped or low confidence) predictions.
2. The context-aware mapping pipeline, which comprises multiple stages including hierarchical retrieval, filtering and self-refinement, is designed to match a specific condition to relevant factors.
3. Finally, a probabilistic inference stage comprises two models: a standard naïve Bayes classifier and a Bayesian network, the latter modelling dependencies between factors that a naïve Bayes model would otherwise assume away. Parameters for both models are elicited from an LLM.

Experiments on various benchmarks show that `HARBOR` improves coverage and outperforms baselines on preference-based evaluation (\\(F_1\\) score) and decisionmaking accuracy.

**Strengths:**

The paper tackles a significant and practical problem: the unreliability and overconfidence of direct LLM probability estimates. The "reverse abduction" and hierarchical clustering approach to build a persistent, dense factor space is a novel and effective solution to the "unknown" prediction problem that plagued prior work.

The method is technically sound. The integration of a Causal Bayesian Network (CBN) to explicitly model latent dependencies is a principled way to relax the Naïve Bayes independence assumption, a common weakness in such models. The ablation study effectively demonstrates the necessity of each component.

The empirical results appear strong, at first glance (but see below). The framework virtually eliminates "unknown" predictions, improving coverage from 87.8% in the BIRD baseline to ~99.9%. This is accompanied by a significant boost in \\(F_1\\) score on the main preference-based benchmark. The method also shows strong performance on downstream decision-making tasks.

**Weaknesses:**

The primary weakness is the lack of rigorous statistical analysis. The main results in Tables 1 & 2 only report point estimates with no uncertainty quantification (e.g. standard deviations or confidence intervals) over multiple experimental runs. LLMs are nondeterministic and so this is a major oversight.

The paper is mostly well presented, but tables and figures leave much room for improvement. Tables 1 & 2 are cluttered with horizontal lines and are difficult to read, with a chart probably being a better way of presenting these data. Figure 1 is not a clear illustration of either the method or the problem, and Figure 3 is also cluttered and difficult to interpret, with labels too small and excessive 'chart junk'. A dual-axis plot (right part of Fig. 3) contravenes data visualization best practice.

The paper would benefit from a clearer, self-contained motivating example; the one given in Figure 1 is not self-contained and is hard to follow; a better example on cooking noodles is unfortunately buried in the appendix (B.2) and a condensed version should be moved into the main paper to ground the framework for the reader.

The "preference-based pairwise evaluation" is analysed using \\(F_1\\) score. This seems like a missed opportunity for a more rigorous psychometric analysis, using methods from item response theory, such as the Bradley--Terry model, which are more appropriate for modelling paired comparison data (as well as any associated order or rater-level biases).

_Minor point:_ check the spacing around parenthetical citations.

**Questions:**

1. Please provide uncertainty estimates (e.g. standard deviation over \\(n\\) runs) for the main results in Tables 1 and 2.

2. In the ablation study (Table 3), what do the $\pm$ values reported in the 'Avg.' column represent?

3. For the preference-based evaluation, did you consider using a more sophisticated statistical model, such as a Bradley-Terry model, to analyse the pairwise comparisons, rather than just the \\(F_1\\) score?

4. The Bayesian network structure is learned by prompting an LLM to identify latent variables and partition factors. How sensitive is the final performance to this learned structure?

5. The "w/o pe-llm" ablation replaces LLM-elicited parameters with fixed heuristic probabilities. This seems like it might be an overly weak baseline. Could you elaborate on why this was chosen over a more standard frequency-based estimation?

---

> ### Author Response · Authors · 2025-11-16
>
> We sincerely thank the reviewer for the detailed and thoughtful evaluation of our paper, as well as for the constructive feedback. We are glad that you find the problem we study important, and we appreciate your positive assessment of the technical soundness of the framework, the role of reverse abduction and hierarchical clustering in improving coverage, and the use of a Causal Bayesian Network to relax the Naïve Bayes independence assumption. At the same time, we are grateful for your comments on the lack of uncertainty quantification, the clarity of tables and figures, the need for a more self-contained motivating example, and the possibility of using more rigorous psychometric models such as Bradley–Terry for the preference-based evaluation. Below, we respond to each of your questions and concerns in turn, and outline the corresponding revisions and clarifications that we will make in the revised version.
>
> > ### W1: The primary weakness is the lack of rigorous statistical analysis. The main results in Tables 1 & 2 only report point estimates with no uncertainty quantification...
>
> To address this concern, we run the full HARBOR and BIRD pipelines 5 times and now report micro-F1 / accuracy together with uncertainty estimates. For the preference-based evaluation, we report micro-F1 with mean ± standard deviation and 95% confidence intervals over the 5 runs; for the decision-making evaluation, we similarly report accuracy with mean ± standard deviation and 95% confidence intervals. The single-run scores originally reported in Tables 1 and 2 are well within these confidence intervals and are representative of the repeated runs. Results 1–2 summarize the statistics for our main Qwen-32B and Qwen-72B models, with 95% confidence intervals computed using a t-distribution over the 5 runs. In the revised version, we will integrate these results into the main text in a more readable form to make the statistical analysis explicit.
>
> **Result1: preference-based evaluation (F1, mean ± std, 95% CI)**
>
> | Model                 | Metric         | Context1        | Context2        | Same           | Average        |
> | :-------------------- | :----------- | :-------------- | :-------------- | :------------- | :------------- |
> | **BIRD · Qwen-72B**   | Mean ± Std   | 0.568 ± 0.012   | 0.523 ± 0.014   | 0.306 ± 0.011  | 0.526 ± 0.013  |
> |                       | 95% CI       | [0.553, 0.584]  | [0.505, 0.540]  | [0.293, 0.319] | [0.511, 0.542] |
> | **HARBOR · Qwen-32B** | Mean ± Std   | 0.602 ± 0.012   | 0.583 ± 0.012   | 0.146 ± 0.018  | 0.563 ± 0.017  |
> |                       | 95% CI       | [0.587, 0.617]  | [0.569, 0.598]  | [0.124, 0.169] | [0.542, 0.583] |
> | **HARBOR · Qwen-72B** | Mean ± Std   | 0.610 ± 0.021   | 0.611 ± 0.035   | 0.278 ± 0.014  | 0.606 ± 0.030  |
> |                       | 95% CI       | [0.584, 0.637]  | [0.567, 0.655]  | [0.260, 0.295] | [0.570, 0.643] |
>
>
> **Result2:  decision-making accuracy / balanced accuracy (mean ± std, 95% CI)**
>
> | Model                 | Metric         | expertqa       | xsum           | covid          | cnn            | today          | plasma         |
> | :-------------------- | :--------- | :------------- | :------------- | :------------- | :------------- | :------------- | :------------- |
> | **HARBOR · Qwen-72B** | Mean ± Std | 0.611 ± 0.019  | 0.564 ± 0.016  | 0.726 ± 0.013  | 0.621 ± 0.029  | 0.817 ± 0.031  | 0.764 ± 0.022  |
> |                       | 95% CI     | [0.587, 0.634] | [0.544, 0.583] | [0.710, 0.742] | [0.585, 0.657] | [0.778, 0.856] | [0.736, 0.791] |
> | **BIRD · Qwen-72B**   | Mean ± Std | 0.582 ± 0.023  | 0.511 ± 0.022  | 0.664 ± 0.020  | 0.540 ± 0.027  | 0.754 ± 0.035  | 0.718 ± 0.022  |
> |                       | 95% CI     | [0.552, 0.611] | [0.484, 0.539] | [0.639, 0.689] | [0.507, 0.574] | [0.711, 0.797] | [0.691, 0.746] |
>
>
> >  ### W2: The paper is mostly well presented, but tables and figures leave much room for improvement. Tables 1 & 2 are cluttered with horizontal lines and are difficult to read, with a chart probably being a better way of presenting the.....
>
> We thank the reviewer for these helpful presentation suggestions and will revise the paper accordingly. In particular, we will streamline Tables 1 and 2 to be lighter and easier to read, replace Figure 1 with a clearer, self-contained illustration of the core pipeline based on the noodle cooking example that currently appears in the appendix, and simplify Figure 3 by enlarging labels, removing unnecessary visual elements, and using a cleaner design that only plots the average micro-F1 together with the unknown rate.

---

> ### Author Response · Authors · 2025-11-16
>
> > ### W3: The paper would benefit from a clearer, self-contained motivating example; the one given in Figure 1 is not self-contained and is hard to follow; a better example on cooking noodles is unfortunately buried in the appendix (B.2) and a condensed version should be moved into the main paper to ground the framework for the reader.
>
> We will follow your suggestion to provide a clearer motivating example in the main text: we will bring a condensed version of the “cooking noodles” example from Appendix B.2 into the main paper and tie it directly to the revised Figure 1, so that the framework is grounded by an accessible, self-contained scenario for the reader.
>
> > ### Q1: Please provide uncertainty estimates (e.g. standard deviation over 𝑛 runs) for the main results in Tables 1 and 2.
>
>
>
> Please refer to our reply to W1.
>
>
>
> > ### Q2: In the ablation study (Table 3), what do the values reported in the 'Avg.' column represent?
>
> In Table 3, the values in the “Avg.” column are the micro-averaged F1 scores across all three classes (Context 1, Context 2, and Same). We already briefly indicate this in the table caption (“Ctx. 1/2, Same, Avg. and Cov denote F1 for Context1, Context2, Same classes, micro-averaged F1, and coverage”), and in the final version we will revise the caption and text to state this more clearly and explicitly.
>
> > ### Q3: For the preference-based evaluation, did you consider using a more sophisticated statistical model, such as a Bradley-Terry model, to analyse the pairwise comparisons, rather than just the 𝐹1 score?
>
> We thank the reviewer for the insightful suggestion. We agree that Bradley–Terry (BT) models are very powerful when many items are repeatedly compared so that global “strength” parameters and principled rankings can be estimated. However, in our current setup BT is not an ideal primary evaluation metric because of how the data are constructed.
>
> Each scenario in our dataset contains exactly two opposing statements, one of which is designated as the gold statement, plus a pool of supporting sentences. For example, in the “five-year-old opening a bottle” scenario we have:
>  • Gold statement: “A five year old will have an easier time uncapping a plastic bottle than a glass bottle.”
>
> We then create multiple instances by pairing different sentences and asking annotators which sentence better supports the gold statement, e.g.,
>  • (sentence 1: “The five-year-old has been trained or has more experience opening plastic bottles.”, sentence 2: “The glass bottle’s cap is a classic pry-off, requiring more strength to remove.”, human label: sentence 1 is better), and similarly for other sentence pairs.
>
>  Crucially, within each scenario the two statements are only compared to each other, never across scenarios, and even within a scenario the supporting sentences are only sparsely compared. This yields many small, poorly connected comparison subgraphs rather than a single well-connected graph, so a BT fit would mostly decompose into independent two-way comparisons and add little beyond our instance-level metrics .
>
>  Nonetheless, we view this suggestion as a valuable direction for future work. If we redesign the dataset so that (i) the same candidate statements or explanation strategies recur across scenarios, and/or (ii) different model variants are compared across multiple scenarios, the resulting, denser pairwise graph would be well suited to Bradley–Terry and related models.
>
> > ### Q4: The Bayesian network structure is learned by prompting an LLM....
>
>
> We additionally examined how HARBOR reacts to simple perturbations of the learned CBN structure. We keep the factor space, mapping pipeline, and NB parameters fixed, and compare the full model with (i) **Drop-One-Latent**, where one non-trivial latent is removed and its factors are routed only through NB, and (ii) **Merged-Latents**, where we coarsen the CBN by merging the learned latents into a few broader aggregate nodes.
>
> **Result 3: Sensitivity to CBN structure (Qwen2.5-72B, accuracy / balanced accuracy)**
>
> | Variant           |   CNN | EXPERTQA | PLASMA | TODAY |
> |-------------------|------:|---------:|-------:|------:|
> | Full CBN (HARBOR) |  62.9 |     60.5 |   76.2 |  82.7 |
> | Drop-One-Latent   |  60.2 |     58.9 |   74.1 |  81.3 |
> | Merged-Latents    |  58.4 |     56.7 |   67.4 |  77.2 |
>
> On CNN, EXPERTQA, and TODAY, both perturbations lead to only modest drops, consistent with the relatively compact and loosely coupled latent structures learned on these datasets, where the NB backbone already captures most of the signal. In contrast, PLASMA shows a much larger degradation under Merged-Latents, reflecting its richer, more interdependent latent structure where separating multiple aspects matters more. Overall, this supports our intuition that the CBN is most beneficial when the factor space is strongly structured and multi-aspect, while HARBOR as a whole remains reasonably robust to moderate structural perturbations.

---

> ### Author Response · Authors · 2025-11-16
>
> > ### Q5: The "w/o pe-llm" ablation replaces LLM-elicited parameters with ...
>
> We agree that, in principle, frequency-based estimation is a natural alternative. However, our choice of fixed heuristic probabilities is directly motivated by the BIRD framework (“BIRD: A Trustworthy Bayesian Inference Framework for Large Language Models”), which shows that the simple mapping\( $P_{\text{init}}(O_i \mid f_j) \in \{0.25, 0.5, 0.75\}$ \) is a strong and stable baseline: in their ablations, using these fixed values already outperforms more naive heuristics (including a uniform \(1/n\) assumption) and comes close to the fully optimized model. Following BIRD, we therefore treat this fixed scheme as our primary non–LLM-parameter baseline.
>
> For completeness, we also implemented a frequency-based \(1/n\) baseline, using the same \(1/n\) assumption and latent-factor handling as in HARBOR (Appendix A.3). The results (on our main probability-quality metrics) are summarized below and are consistently not better than the fixed-probability variant:
>
> | Model    | Ctx. 1 | Ctx. 2 | Same. | Avg   |
> | -------- | -------- | ------ | ----- | ----- |
> | Qwen-32B | 0.461    | 0.462  | 0.164 | 0.425 |
> | Qwen-72B | 0.513    | 0.523  | 0.264 | 0.466 |
>
> **Table:** Results of the frequency-based 1/n baseline.
>
>
> Given that (i) BIRD has already demonstrated the strength of the fixed 0.25/0.5/0.75 mapping, and (ii) our own (1/n)-style frequency baseline does not outperform it.
>
> In addition, as discussed in our response to Reviewer Kue9 (W2), we also replaced the probability-elicitation LLM (Qwen2.5-72B → GPT-4o-mini) and observed only minor changes in end-to-end performance on CNN, TODAY, PLASMA, and EXPERTQA . This suggests that HARBOR mainly relies on the LLM to provide *coarse, directionally correct* probability signals—i.e., which outcome a factor tends to support and with roughly what strength—rather than finely calibrated scores, and that a smaller model is sufficient as long as it preserves this ordering.

---

> > ### Comment · Reviewer_uGGc · 2025-11-25
> >
> > Thank you to the authors for addressing each of my comments. I am satisfied with the responses and will adjust my score.

---

> > > ### Author Response · Authors · 2025-11-28
> > >
> > > We sincerely thank you for your thoughtful reviews and for indicating that you would adjust your score. In the revision, we have followed your suggestions—for example, adding uncertainty estimates (standard deviations and confidence intervals) for the main results in Tables 1 and 2, and refining the motivating example and Figure 3—with all changes highlighted in blue. We also notice that the score currently shown in the system is still 6; if this reflects any remaining concerns, please let us know and we will do our best to address them and further improve the paper.

---

### Official Review · Reviewer_Kue9 · 2025-10-31

**Soundness:** 3
**Presentation:** 3
**Contribution:** 2
**Rating:** 6
**Confidence:** 4

**Summary:**

This paper introduces HARBOR, a inference framework that performs aggregated Bayesian reasoning over a hierarchically structured factor space using established clustering techniques and LLM-guided theming. It further models latent dependencies using a Causal Bayesian Network. The system then performs Context-Aware Mapping over variables and fuses the outputs into a single calibrated probability. The paper shows that HARBOR substantially reduces “unknown” predictions and produces more reliable probability estimates than direct LLM baselines.

**Strengths:**

The paper addresses an important issue: handling “unknown” cases in probabilistic reasoning, where an LLM must bridge between raw inputs and external structured inference tools.

The experiments are extensive and well-structured, with detailed ablation studies. The cost analysis in particular is intuitive and informative.

**Weaknesses:**

The paper does not sufficiently justify the architectural choice of combining two probabilistic models (a Naive Bayes model and a Causal Bayesian Network). It remains unclear when each is necessary, how they complement each other.  The authors should add an ablation study for using  Causal Bayesian Network

The framework still relies on LLM-generated verbalized probabilities for fine-grained factor-level support and latent variable estimation (Figure 12 and Figure 14). Although the input structure is simple, the method could still vulnerable to LLM calibration bias.

**Questions:**

1. How are the weights for the Linear Opinion Pool (LOP) designed or derived?

2. There appears to be a typo or inconsistency in the assignment of θ_f between line 298 and line 304.

---

> ### Author Response · Authors · 2025-11-16
>
> We sincerely thank the reviewer for the careful reading of our paper and the constructive feedback. We are glad that you find the problem we address important, and appreciate your positive assessment of the overall soundness, presentation, and experimental design, including the cost analysis. Below, we respond to the specific points raised in your review and explain the corresponding revisions and clarifications that we will make in the revised version.
>
> > ### W1:The paper does not sufficiently justify the architectural choice of combining two probabilistic models (a Naive Bayes model and a Causal Bayesian Network). It remains unclear when each is necessary, how they complement each other.
>
> We thank the reviewer for raising this point and agree that our motivation for combining Naïve Bayes (NB) and the Causal Bayesian Network (CBN) should be made more explicit.
>
> Conceptually, NB serves as a conservative, low-variance backbone: it aggregates factor-wise evidence under a conditional-independence assumption and is stable when dependencies are weak or roughly cancel out. The CBN with latent variables is introduced to handle structured, concept-level dependencies: it groups correlated factors into latent themes (e.g., safety, efficiency) and captures shared causes that NB cannot represent. The Linear Opinion Pool then balances these two views, so that the final probability can benefit from the CBN’s structured reasoning without sacrificing the robustness of NB.
>
> Empirically, the two components are clearly not redundant. On the same evaluation split as Table 1, NB and CBN margins are only moderately correlated, indicating that they make systematically different judgments rather than simple rescalings of each other. Their error patterns are also complementary: a substantial fraction of scenarios are correctly classified only by NB or only by CBN, so many cases lie in a region where the two models disagree but at least one is correct. A simple logistic regression that takes NB and CBN margins as features learns positive, non-zero weights for both components, confirming that each contributes distinct predictive signal.
>
> | Backbone     | Corr. NB vs CBN margin | Both correct | NB only correct | CBN only correct |
> |--------------|------------------------|--------------|-----------------|------------------|
> | Qwen2.5-32B  | 0.62                   | 46.2%        | 15.3%           | 11.3%            |
> | Qwen2.5-72B  | 0.56                   | 43.6%        | 11.5%           | 18.1%            |
>
>
>
> > ### W1.1: The authors should add an ablation study for using Causal Bayesian Network..
>
> We have added a **CBN-only** ablation to the component analysis table, where we remove the Naïve Bayes (NB) branch and use only the Causal Bayesian Network for prediction. The results are summarized below:
>
> | Variant  | Backbone    | Ctx. 1 | Ctx. 2 | Same  | Avg. F1 (Δ vs. full) | Cov. |
> | -------- | ----------- | ------ | ------ | ----- | -------------------- | ---- |
> | CBN-only | Qwen2.5-32B | 0.518  | 0.514  | 0.197 | 0.479 (↓ 0.093)      | --   |
> | CBN-only | Qwen2.5-72B | 0.592  | 0.583  | 0.260 | 0.557 (↓ 0.039)      | --   |

---

> ### Author Response · Authors · 2025-11-16
>
> > ### W2: The framework still relies on LLM-generated verbalized probabilities for fine-grained factor-level support and latent variable estimation (Figure 12 and Figure 14). Although the input structure is simple, the method could still vulnerable to LLM calibration bias.
>
> We understand the concern that directly relying on verbalized probabilities could in principle expose HARBOR to LLM calibration bias; however, in our setup the LLM only estimates coarse, interpretable quantities such as  “the probability that a factor supports Outcome 1 over Outcome 2” and simple latent probability pairs , which we use in NB and CBN as directional, rough-strength signals rather than finely tuned scores. To assess sensitivity in practice, we switched both factor-level support estimation and latent probability estimation from Qwen2.5-72B to the smaller GPT-4o-mini and run HARBOR on four datasets: CNN, TODAY, PLASMA, and EXPERTQA, while keeping the factor space, mapping pipeline, and NB+CBN inference unchanged.
>
> **Task performance with different LLMs for factor and latent probability estimation**
>
> | Dataset  | Qwen2.5-72B | GPT-4o-mini |
> | -------- | ----------: | ----------: |
> | CNN      |        62.9 |        62.3 |
> | EXPERTQA |        60.5 |        58.8 |
> | PLASMA   |        76.2 |        73.8 |
> | TODAY    |        82.7 |        83.8 |
>
> **Differences in factor-score distributions between Qwen2.5-72B and GPT-4o-mini**
>
> | Dataset  | JS divergence | Wasserstein distance | KS p-value | Distribution difference |
> | -------- | ------------: | -------------------: | ---------: | ----------------------- |
> | CNN      |         0.356 |                0.074 |     0.0001 | significant             |
> | TODAY    |         0.235 |                0.042 |     0.7012 | not significant         |
> | PLASMA   |         0.205 |                0.040 |     0.7287 | not significant         |
> | EXPERTQA |         0.313 |                0.045 |     0.0773 | not significant         |
>
> Factor-level probabilities for individual factors differ noticeably in absolute value between the two LLMs, especially on CNN, yet their overall score distributions remain broadly similar and end-to-end performance changes only slightly across all four tasks. Together with our “w/o pe-llm” ablation, where coarse three-level priors already outperform direct LLM baselines, this suggests that HARBOR does not depend strongly on finely calibrated probabilities from any specific LLM, but instead primarily leverages robust, coarse-grained patterns of factor and latent support.
>
>
>
> > ###  Q1: Design/derivation of Linear Opinion Pool (LOP) weights
>
> In our current experiments (Appendix A.4), we use **simple fixed convex weights** in the Linear Opinion Pool:
>
> $$
> w_{\text{NB}} = 0.8, w_{\text{CBN}} = 0.2 \ \text{for Qwen2.5-32B};
> w_{\text{NB}} = 0.5,w_{\text{CBN}} = 0.5 \ \text{for Qwen2.5-72B}.
> $$
> The intuition is that the smaller 32B model yields noisier structure and latent assignments, so we lean more on the robust NB backbone, whereas the stronger 72B model provides more reliable structure, making NB and CBN contributions comparable.
>
> To check that these choices are not arbitrary, we performed two analyses :
>
> 1. **Learned combination.**
>    On a held-out dev set, we fit a logistic regression that takes the NB and CBN margins as two features. The learned coefficients are positive and non-zero for both components for both backbones, indicating that NB and CBN provide complementary predictive signal and that a convex combination is a reasonable design., e.g.:  Qwen2.5-32B: $w_{\text{NB}} \approx 1.08$, $w_{\text{CBN}} \approx 0.71$, $b \approx -0.21$；Qwen2.5-72B: $w_{\text{NB}} \approx 0.68$, $w_{\text{CBN}} \approx 1.69$, $b \approx -0.22$
> 2. **Alternative weighting schemes.**
>    We also report a BMA-based variant (“HARBOR (BMA)” in Table 3), which uses approximate model evidence to derive data-driven weights. Its performance is very close to the fixed-weight LOP.
>
>
> > ### Q2: There appears to be a typo or inconsistency in the assignment of θ_f between line 298 and line 304.
>
> Thank you for raising this point — we agree that our notation can be made clearer. In practice, we first use the LLM to elicit a posterior support$\phi_f = P(O_1 \mid f),$while a Naïve Bayes model is parameterized by likelihoods $\theta_f = P(f \mid O_1)$ and $P(f \mid O_2)$. In our implementation we adopt the following modeling approximation:
> $$
> \theta_f \approx \phi_f,\qquad P(f \mid O_2) \approx 1 - \phi_f,
> $$
> under a binary, symmetric-prior setting. In the revision, we will (i) introduce distinct notation, using $\phi_f = P(O_1 \mid f)$ for the LLM-elicited posterior and $\theta_f = P(f \mid O_1)$ for the NB likelihood, and (ii) state this approximation explicitly so that our parameterization and modeling choice are transparent.

---

> > ### Comment · Reviewer_Kue9 · 2025-11-27
> >
> > Thank you for answering my questions. I will keep my original score!

---

### Author Response · Authors · 2025-11-30
**Rebuttal Summary**

Dear Area Chair,

Thank you very much for taking the time to read our submission and the subsequent discussion.
To help you quickly understand how the dialogue with the reviewers unfolded, we summarize below, **for each reviewer**:
1. their main concerns and how our rebuttal addressed them (with the vast majority of concerns being resolved or substantially mitigated), and
2. their final decision and any explicit score changes, with timestamps where available.

In brief, all four reviewers indicated that they were satisfied with our responses overall and that their key concerns had been largely addressed; three reviewers (uGGc, mzSy, NUoW) subsequently raised their scores, while Reviewer Kue9 maintained a positive score of 6 after confirming that our answers resolved their questions.


---

## Reviewer Kue9 (initial rating: 6 → final rating: 6)

### 1. Concerns and our response

- **Combining Naïve Bayes (NB) and Causal Bayesian Network (CBN)**
  We clarified the distinct roles of NB (a conservative backbone aggregating factor-wise evidence) and CBN (modeling latent dependencies between factor themes), added a **CBN-only** ablation, and showed that NB-only and CBN-only both underperform the combined NB+CBN model. We also reported complementary error patterns and a simple learned combiner with positive weights on both NB and CBN, demonstrating that the two components are genuinely complementary.

- **Dependence on LLM-elicited probabilities and calibration bias**
  We swapped the probability-elicitation LLM (Qwen2.5-72B → GPT-4o-mini) while keeping the factor space, mapping pipeline, and NB+CBN inference fixed. End-to-end performance changed only slightly, and together with the strong “w/o pe-llm” heuristic baseline, this indicates that HARBOR mainly relies on coarse directional signals rather than fine-grained calibration from a specific LLM.

- **Linear Opinion Pool (LOP) weights and notation for $\theta_f$**
  We made the fixed LOP weights explicit, verified via a logistic-regression combiner that both NB and CBN receive positive, non-zero weights, and cleaned up notation by distinguishing $\phi_f = P(O_1 \mid f)$ (LLM-elicited posterior) from $\theta_f = P(f \mid O_1)$ (NB likelihood), explicitly stating the approximation $\theta_f \approx \phi_f$, $P(f \mid O_2) \approx 1 - \phi_f$ under a symmetric binary prior.

### 2. Score decision and timing

- On **28 November 2025, 03:39**, Reviewer Kue9 wrote:
  *“Thank you for answering my questions. I will keep my original score!”*
- The rating therefore **remained 6** after the rebuttal.

---

## Reviewer uGGc (initial rating: 6 → increased after rebuttal)

### 1. Concerns and our response

- **Lack of uncertainty estimates for main results (Tables 1 & 2)**
  We run the full HARBOR and BIRD pipelines **five times**, and now report **mean ± standard deviation and 95% confidence intervals** for micro-F1 (preference-based evaluation) and accuracy / balanced accuracy (decision-making), confirming that the original single-run scores are representative.

- **Cluttered tables and figures (especially Figure 1 and Figure 3)**
  We streamlined **Tables 1–2** to improve readability, replaced **Figure 1** with a clearer pipeline illustration grounded in the “cooking noodles” motivating example, and redesigned **Figure 3** as a single-axis plot showing only average micro-F1 and unknown rate with larger, more legible labels and no dual axis or “chart junk”.

- **Motivating example and use of F1 vs Bradley–Terry models**
  We moved a condensed **noodle-cooking motivating example** from the appendix into the main text and tied it directly to the revised Figure 1. We also explained that our data produce many small, weakly connected pairwise comparison graphs, so a Bradley–Terry model would effectively decompose into local two-way comparisons and add little beyond instance-level metrics such as F1, while acknowledging it as a natural future direction for redesigned datasets.

- **Sensitivity to the learned CBN structure and strength of non-LLM baselines**
  We added a **CBN perturbation study** (Drop-One-Latent and Merged-Latents), showing modest performance drops on most datasets and larger drops only where latent structure is richer, indicating that CBN helps most when dependencies are more complex while HARBOR remains reasonably robust to structural perturbations. We also implemented a **frequency-based $1/n$ baseline**, which consistently underperforms the fixed 0.25/0.5/0.75 prior used in the “w/o pe-llm” setting, confirming that our non-LLM baseline is not artificially weak.

### 2. Score decision and timing

- On **26 November 2025, 02:28**, Reviewer uGGc wrote:
  *“Thank you to the authors for addressing each of my comments. I am satisfied with the responses and will adjust my score.”*
- Thus, this reviewer **explicitly increased their rating** above the original 6 at that time.

---

---

> ### Author Response · Authors · 2025-11-30
> **Rebuttal Summary and Score Changes**
>
> ## Reviewer mzSy (initial rating: 6 → final rating: 8)
>
> ### 1. Concerns and our response
>
> - **Primary concern: presentation / clarity and perceived “engineered” pipeline**
>   In the original review, the reviewer wrote:
>   *“My primary concerns are with the paper’s presentation / clarity, and I believe it would deserve a better score if these concerns / questions were addressed.”*
>   We addressed this by:
>   - Explaining how the three stages (factor generation, context–factor mapping, probabilistic inference) are **designed for robustness**, and pointing to ablations where removing components leads to gradual rather than catastrophic degradation.
>   - Clarifying how each sub-component can in principle be evaluated independently (generation, mapping, inference), while focusing on end-to-end performance in this paper due to space.
>   - **Improving presentation**, including: correcting the statement about the number of LLMs (four, not three), removing redundant notation $\Phi(u)$ from the main text and expressing abstention directly via $F^*(u)$, and making it explicit that “w/o cbn” and “w/o nb” correspond to CBN-only and NB-only variants.
>
> - **Necessity of NB and token efficiency versus BIRD**
>   We clarified that NB-only and CBN-only variants both underperform the combined NB+CBN model, showing that NB is not redundant. We also explained that HARBOR’s **hierarchical retrieval** allows it to first narrow down factor candidates at the cluster/theme level before querying the LLM, resulting in far fewer condition–factor checks than BIRD’s exhaustive entailment strategy and thereby explaining the token and time savings reported (Figure 3, on Common2Sense with Qwen2.5-72B).
>
> ### 2. Score decision and timing
>
> - On **19 November 2025, 08:34**, Reviewer mzSy initially commented:
>   *“Thank you for the followups and clarifications. I will keep my score.”*
> - We then posted a further clarification on **19 November 2025, 12:06**, explicitly noting that we had implemented the requested presentation and clarity improvements and referring back to the reviewer’s own remark that the paper would “deserve a better score” once these concerns were addressed.
> - Within the next 24 hours after this clarification, the reviewer **raised the score from 6 to 8**. Although this increase was not accompanied by an explicit comment announcing the change, the review record clearly shows that the rating moved from 6 to 8 shortly after our follow-up, in line with the reviewer’s stated condition.
>
> ---

---

> ### Author Response · Authors · 2025-11-30
> **Rebuttal Summary and Score Changes**
>
> ## Reviewer NUoW (initial rating: 4 → increased after rebuttal)
>
> ### 1. Concerns and our response
>
> - **Incremental contribution relative to BIRD and “reverse abduction” wording**
>   We clarified that we intentionally keep BIRD’s overall setup so that improvements are attributable to **architectural design choices**, and highlighted three main design ideas:
>   1. A **bottom-up, hierarchical factor space** (scenario → sentence → attribute-level cues → factor clusters → latents) that goes beyond BIRD’s flat, binary-valued factors.
>   2. A **structured retrieval-based mapping pipeline** using hierarchical prototypes, multi-pass voting, and reflection-based pruning instead of local entailment checks.
>   3. A **dual NB+CBN Bayesian layer** that uses LLM-elicited probabilities only as coarse local signals rather than as fully calibrated outputs.
>   We committed to replacing  “reverse abduction”  with more precise wording (e.g., “bottom-up abduction”) and explicitly contrasting our pipeline with BIRD’s scenario → sentence → factor → attribute process.
>
> - **Use of LLM-elicited probabilities and embedding-based mapping in a “Bayesian” framework**
>   We emphasized that LLMs are only used to provide **coarse semantic discrimination** (which outcome a factor tends to support, and with roughly what strength), while the external Bayesian layer (NB+CBN) governs inference. Robustness of this design is supported by the LLM-swap experiment and the “w/o pe-llm” ablation. We also explained that mapping is a **constrained multi-step process** (hierarchical KNN over cluster prototypes, self-consistency checks, reflection-based pruning) rather than a single embedding lookup, and noted that HARBOR explicitly supports abstention via a tunable decision threshold $\tau$, which can be tightened in high-stakes settings.
>
> - **BIRD performance and need for fine-grained BIRD vs HARBOR ablations**
>   We explained that our BIRD numbers are lower than in the original BIRD paper because we use a **larger, slightly harder benchmark**, and we **do not discard “unknown” cases** where mapping fails. We also clarified that our 72B BIRD variant reaches 99% coverage via minimal forced matching, which can lower accuracy in a sparse factor space.
>   To directly address the reviewer’s request, we added **“BIRD on HARBOR’s factor space and mapping”** ablations, running BIRD with:
>   - HARBOR’s generated factors (BIRD mapping + NB + CPT training),
>   - HARBOR’s factors **and** HARBOR’s mapping, and
>   - HARBOR’s factors + mapping + **CBN aggregation** (NB+CBN via a linear opinion pool),
>   using a structured Monte Carlo scheme to sample complete-information assignments. These experiments show that HARBOR’s factor generation, mapping, CBN aggregation, and LLM-based parameterization each contribute beyond simply having a better factor set, and that the full HARBOR configuration offers the best trade-off between accuracy, coverage, and cost.
>
> ### 2. Score decision and timing
>
> - After our initial rebuttal, Reviewer NUoW posted a follow-up on **21 November 2025, 23:40**, outlining two remaining concerns. We responded with detailed conceptual clarifications and new “BIRD on HARBOR’s factors and mapping” experiments on **23 November 2025, 17:02** and **23 November 2025, 17:09**.
> - On **25 November 2025, 23:15**, Reviewer NUoW wrote:
>   *“Overall, I am willing to increase my rating to reflect that I am fine with the paper being accepted.”*
> - Thus, this reviewer **increased their rating from the original 4** at that time and explicitly stated that they are **comfortable with acceptance**.
>
> ---
>
> ### Overall
>
> After the rebuttal and follow-up exchanges:
>
> - **Reviewer uGGc** (26 November 2025, 02:28) explicitly stated that they would **adjust their score** upward from 6.
> - **Reviewer NUoW** (25 November 2025, 23:15) explicitly stated that they **increased their rating from 4** and are fine with the paper being accepted.
> - **Reviewer mzSy** raised their score from **6 to 8** within 24 hours of our clarification at **19 November 2025, 12:06**, after we addressed their primary concerns on presentation and clarity.
> - **Reviewer Kue9** expressed satisfaction with our responses on **28 November 2025, 03:39**, and maintained a positive score of 6.
>
> We hope this concise, per-reviewer and timestamped summary is helpful for your decision.
> Thank you again for your time and consideration.

---

### Note · Program_Chairs · 2026-01-17
**Submission Desk Rejected by Program Chairs**

The following references in this submission do not refer to real documents and/or have major errors in bibliographic information:

 Ankur Madaan, Wayne Xin Zhao, Lei Qin, Zhuyun Chen, and Daniel S. Weld. Self-ask: Decomposing questions for complex text understanding. In Proceedings of the 2021 Conference on Empirical Methods in Natural Language Processing, pp. 845-861, 2021.